# *Streptococcus pneumoniae* triggers hierarchical autophagy through reprogramming of LAPosome-like vesicles via NDP52-delocalization

Michinaga Ogawa [1,11]*, Naoki Takada[1,2,11], Sayaka Shizukuishi[1,3], Mikado Tomokiyo[1,4], Bin Chang[1], Mitsutaka Yoshida[5], Soichiro Kakuta[5,6], Isei Tanida[6], Akihide Ryo[3], Jun-Lin Guan[7], Haruko Takeyama[2,8,9,10] & Makoto Ohnishi[1]

In innate immunity, multiple autophagic processes eliminate intracellular pathogens, but it remains unclear whether noncanonical autophagy and xenophagy are coordinated, and whether they occur concomitantly or sequentially. Here, we show that *Streptococcus pneumoniae*, a causative of invasive pneumococcal disease, can trigger FIP200-, PI3P-, and ROS-independent pneumococcus-containing LC3-associated phagosome (LAPosome)-like vacuoles (PcLVs) in an early stage of infection, and that PcLVs are indispensable for subsequent formation of bactericidal pneumococcus-containing autophagic vacuoles (PcAVs). Specifically, we identified LC3- and NDP52-delocalized PcLV, which are intermediates between PcLV and PcAV. Atg14L, Beclin1, and FIP200 were responsible for delocalizing LC3 and NDP52 from PcLVs. Thus, multiple noncanonical and canonical autophagic processes are deployed sequentially against intracellular *S. pneumoniae*. The Atg16L1 WD domain, p62, NDP52, and poly-Ub contributed to PcLV formation. These findings reveal a previously unidentified hierarchical autophagy mechanism during bactericidal xenophagy against intracellular bacterial pathogens, and should improve our ability to control life-threating pneumococcal diseases.

[1] Department of Bacteriology I, National Institute of Infectious Diseases, 1-23-1, Toyama, Shinjuku-ku, Tokyo 162-8640, Japan. [2] Department of Life Science and Medical Bioscience, Waseda University, 2-2 Wakamatsu-cho, Shinjuku-ku, Tokyo 162-8480, Japan. [3] Department of Microbiology, Yokohama City University Graduate School of Medicine, 3-9 Fukuura, Kanazawa-ku, Yokohama-shi, Kanagawa 236-0004, Japan. [4] School of Veterinary Medicine, Azabu University, Fuchinobe, Sagamihara-shi, Kanagawa 229-8501, Japan. [5] Laboratory of Morphology and Image Analysis, Research Support Center, Juntendo University, 2-1-1, Hongo, Bunkyo-ku, Tokyo 113-8421, Japan. [6] Department of Cellular and Molecular Neuropathology, Graduate School of Medicine, Juntendo University, 2-1-1, Hongo, Bunkyo-ku, Tokyo 113-8421, Japan. [7] Department of Cancer Biology, University of Cincinnati College of Medicine, CARE/Crawley Building, Suite E-870 3230 Eden Avenue, Cincinnati, OH 45267, USA. [8] Computational Bio Big-Data Open Innovation Laboratory, AIST-Waseda University, 3-4-1 Okubo, Shinjuku-ku, Tokyo 169–0072, Japan. [9] Research Organization for Nano & Life Innovation, Waseda University, 513 Wasedatsurumaki-cho, Shinjuku-ku, Tokyo 162–0041, Japan. [10] Institute for Advanced Research of Biosystem Dynamics, Waseda Research Institute for Science and Engineering, Graduate School of Advanced Science and Engineering, Waseda University, 3-4-1 Okubo, Shinjuku-ku, Tokyo 169–8555, Japan. [11]These authors contributed equally: Michinaga Ogawa, Naoki Takada. *email: micogawa@nih.go.jp

Macroautophagy, hereafter referred to as autophagy, is an evolutionarily conserved catabolic pathway in eukaryotic cells that recycles intracellular organelles under starvation conditions and also degrades damaged organelles and misfolded protein aggregates[1]. Xenophagy, the selective targeting of autophagy to intracellular pathogens, plays a role as a protective cytosolic 'executioner' in the innate immune system, and thus provides the first line of defense against bacterial invaders[2]. During xenophagy, interactions between poly-ubiquitin (Ub) and several distinct autophagy adapters, including p62, NDP52, NBR1, TAX1BP1, and OPTN, provides a scaffold for initiating LC3-lipidation and phagophore formation in the vicinity of pathogen-containing vacuoles or the surface of cytosolic intruders, leading to their engulfment by autophagosomal membranes[2–6]. Lipidated LC3 is also recruited into single-membrane nonautophagosomal compartments. This noncanonical autophagic pathway includes LC3-associated phagocytosis (LAP) in phagocytes that engulf bacteria, fungi, or dead cells[7–11]. In LAP, the upstream regulators of canonical autophagy, such as ULK1/2 and FIP200, are dispensable, but a distinct Beclin1–Vps34 complex lacking Atg14L is required. Furthermore, LAP requires NADPH oxidase-derived reactive oxygen species (ROS) and PI3P generated by the Rubicon/UVRAG–Beclin1–Vps34 complex. There is also some evidence that the Atg16L1 WD domain, which binds poly-Ub chains, is involved in the LAP-like autophagy process induced by monensin and entosis[12].

*Streptococcus pneumoniae* is an infectious pathogen responsible for millions of deaths around the world. This pathogen colonizes the nasopharyngeal epithelia asymptomatically, but can often migrate to sterile tissues and cause life-threatening invasive infections (invasive pneumococcal disease: IPD)[13].

In a previous study, we demonstrated that intracellular *S. pneumoniae* is subject to bactericidal xenophagy mediated by pneumococcus-containing autophagic vacuoles (PcAVs) at 2 h post infection (p.i.)[14]. However, it remains unclear whether intracellular *S. pneumoniae* can trigger LAP or LAP-like autophagy process or not.

In this study, we demonstrated that *S. pneumoniae* can trigger the formation of pneumococcus-containing LC3-associated phagosome (LAPosome)-like vacuoles (PcLVs) and revealed that noncanonical and canonical autophagic processes are deployed sequentially against intracellular bacteria.

## Results

### *Streptococcus pneumoniae* is engulfed in FIP200-, PI3P-, ROS-independent LAPosome-like vacuoles during early stage infection.

We have previously reported that *S. pneumoniae* strain R6 is entrapped by bactericidal PcAVs at 2 h p.i.[14]. LAP-like LC3 lipidation also occurs during the pathogen invasion process in nonphagocytic cells[15,16]. Therefore, we investigated whether *S. pneumoniae* triggers an LAP-like autophagy process during early stage infection in nonmyeloid cells. In these experiments, we used WT, FIP200 knockout, and ULK1/2 double knockout (DKO) MEFs stably expressing GFP-LC3. When the cells were infected with *S. pneumoniae* strain R6 for 1 or 2 h, FIP200- and ULK1/2-independent LC3 recruitment to PcVs (pneumococcus-containing vacuoles) was observed at 1 h p.i. to the same level of WT, and, however, it robustly decreased at 2 h p.i. (Fig. 1a, b and Supplementary Fig. 1A). LC3 recruitment was not observed in Atg5 KO MEFs. Hereafter, we refer to the LC3-associated phagosome (LAPosome)-like autophagic bodies triggered by *S. pneumoniae* at 1 h p.i. as PcLVs (pneumococcus-containing LAPosome-like vacuoles). In electron micrographs of PcLVs, single-membraned pneumococci-engulfing ultrastructures were observed (Fig. 1c), which was distinct from double-membraned

PcAV in FIP200 WT MEFs (Fig. 1c). Consistent with a previous report[7], we found that PcLV formation was not affected by Atg14L depletion (Fig. 1d and Supplementary Fig. 1B).

Next, we investigated the regulation of PcLVs by comparing them with LAPosomes in phagocytes. Previous studies showed that PI3P generated by the Rubicon–Beclin1–Vps34 complex is required for LAPosome formation[7]. However, 3-MA (3-methyladenine, a class III PI3K inhibitor) did not affect PcLV formation (Fig. 1e and Supplementary Fig. 1F), and Rubicon, UVRAG, and WIPI2 were not colocalized with PcLVs (Supplementary Fig. 1G), suggesting that PI3P is not required for PcLV formation. In FIP200 KO MEFs expressing GFP-LC3 G120A, which is defective in LC3 lipidation, recruitment of LC3 to *S. pneumoniae* was abolished (Fig. 1f and Supplementary Fig. 1H), suggesting that LC3-lipidation is required for PcLV formation. A previous study showed that ROS generated by Nox2-based NADPH oxidase plays a pivotal role in LAPosome formation in phagocytes[7]. Treatment with antioxidants such as apocynin, NAC, or GSH ethyl ester did not inhibit PcLV formation (Fig. 1g). Consistent with this, NOX1 KO, NOX4 KO, and p22phox knockdown had no obvious effect on PcLV formation (Fig. 1h and Supplementary Fig. 1C, I, and J). Nox2 expression was not detected in FIP200 KO MEFs (Supplementary Fig. 1D). These results suggested that ROS generated by NADPH oxidase is not required for PcLV formation. Furthermore, TLR2 depletion had no significant effect on PcLV formation (Fig. 1h and Supplementary Fig. 1E and J). Together, these results clearly showed that PcLVs have noncanonical LAPosome features, such as ROS-, PI3P-, and TLR2-independence, although they are somewhat similar to LAPosomes in the sense that they are FIP200-independent and have a single-membraned structure.

Previously, we reported that PcAV formation 2 h p.i. is dependent on membrane rupture by Ply, a pore-forming cytolysin of *S. pneumoniae*[14]. PcLV formation is also dependent on Ply (Fig. 1i and Supplementary Fig. 1K). We found that p62 and poly-Ub to PcLVs were recruited to PcLVs only in the presence of Ply (Fig. 1i, and Supplementary Fig. 1L and M). The level of LC3-II was dramatically increased by infection with WT *S. pneumoniae*, but not the Δ*ply* mutant (Fig. 1j). Notably, the level of p62, a reporter of autophagic flux, was not reduced in cells infected with WT bacteria (Fig. 1j), indicating that PcLVs are nondegradative due to their leaky structure (Fig. 1c). To clarify the involvement of Ply cytolytic activity in PcLV formation, we prepared *S. pneumoniae* harboring Ply with low-cytolytic activity. The *ply* gene exists in a number of allelic forms, and allele 5 Ply derived from serotype 1 and 8 pneumococci has low-cytolytic activity (Supplementary Fig. 1N)[17]. When we compared PcLV-triggering capacities between R6 Δ*ply*:allele 1 *ply* (complemented by R6 original *ply*) and R6 Δ*ply*:allele 5 *ply* (complemented by low-cytolytic *ply*), we found that the cytolytic activity of Ply was essential for induction of PcLVs (Fig. 1k). Next, we investigated whether PcLVs have a bactericidal effect by examining their acidification. Consistent with the results in WT MEFs in our previous study[14], lysotracker staining revealed that PcLVs were not acidified, although vacuoles containing Δ*ply* bacteria were frequently acidified (Fig. 1l, and Supplementary Fig. 1O and P). Finally, we investigated the intracellular survivability of WT *S. pneumoniae* in FIP200 KO MEFs (PcAV-defective but PcLV-permissive) and Atg5 KO MEFs (both PcAV- and PcLV-defective) at 1 h p.i. (Supplementary Fig. 1A). We found that intracellular survivability of *S. pneumoniae* was not affected regardless of the presence of LAP activity (Fig. 1m). Similarly, LAP activity had no effect on pneumococcal invasiveness in MEFs (Supplementary Fig. 1Q). As we reported previously, a 1-h incubation is long enough to evaluate endo-lysosomal bactericidal activity against intracellular *S. pneumoniae*[14]. We confirmed that

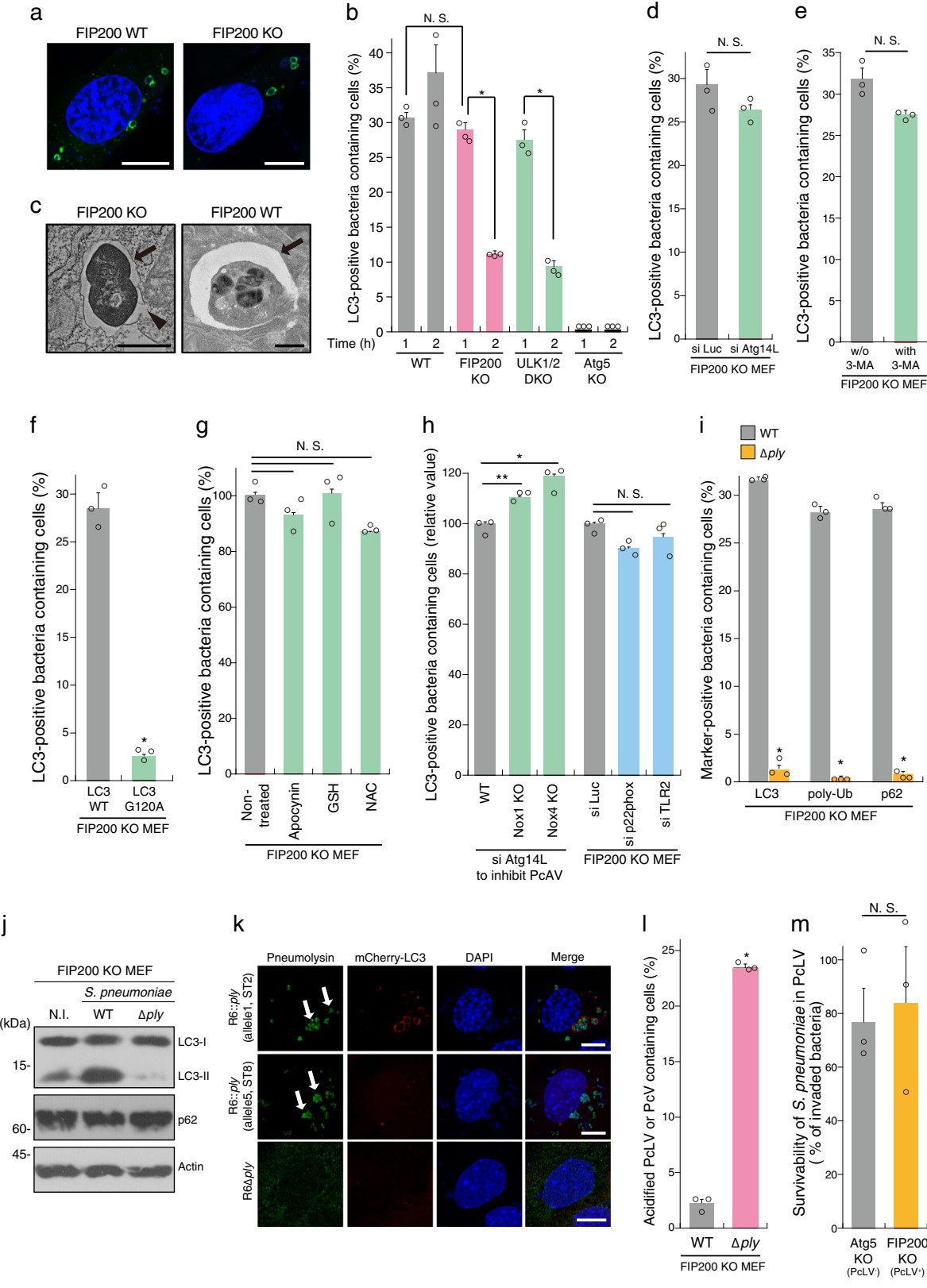

10 µg/ml of penicillin G treatment for 15 min had no negative effect on intracellular survivability of pneumococci until 3 h, and that each cell was invaded by at least one bacterium (Supplementary Fig. 1R). Putting all these results together, we concluded that PcLVs do not have bactericidal activity.

**Noncanonical and canonical autophagic processes are deployed sequentially against intracellular pneumococci.** Multiple autophagic processes, acting simultaneously or sequentially, are involved in targeting intracellular bacterial pathogens[11,15]. To determine whether the autophagic processes involving PcLV and

**Fig. 1 *Streptococcus pneumoniae* is engulfed in FIP200-, PI3P-, ROS-independent LAPosome-like vacuoles during early stage of infection. a** Indicated MEFs/GFP-LC3 infected with pneumococci for 1 h were stained with DAPI. **b** Indicated MEFs/GFP-LC3 infected with pneumococci for 1 or 2 h and stained with DAPI, and percentages of PcLV-containing cells were quantified. **c** Micrographs of FIP200 KO MEFs at 1 h p.i. or FIP200 WT at 2 h p.i.; Bar, 1 μm. Arrows indicate PcLV or PcAV, and arrowhead indicates ruptured PcLV. **d** Indicated MEFs/GFP-LC3 treated with indicated siRNAs were infected with pneumococci for 1 h and stained with DAPI, and percentages of PcLV-containing cells were quantified. **e** FIP200 KO MEFs/GFP-LC3 infected with pneumococci for 1 h with or without 3-methyladenine were stained with DAPI, and percentages of PcLV-containing cells were quantified. **f** FIP200 KO MEFs/GFP-LC3 G120A infected with pneumococci for 1 h were stained with DAPI, and percentages of PcLV-containing cells were quantified. **g** FIP200 KO MEFs/GFP-LC3 infected with pneumococci for 1 h with or without indicated antioxidants were stained with DAPI, and percentages of PcLV-containing cells were quantified. **h** Indicated MEFs/GFP-LC3 treated with indicated siRNA were infected with pneumococci for 1 h and stained with DAPI, and percentages of PcLV-containing cells were quantified. **i** FIP200 KO MEFs/GFP-LC3 infected with indicated pneumococcal strains for 1 h were stained with DAPI or antibodies against pneumococci, and anti-poly-Ub or -p62 antibodies, and percentages of LC3-, poly-Ub-, or p62-positive bacteria containing cells were quantified. **j** Lysates from FIP200 KO MEFs infected with indicated pneumococcal strains for 1 h were subjected to immunoblotting with indicated antibodies. **k** FIP200 KO MEFs/mCherry-LC3 infected with indicated pneumococcal strains for 1 h were stained with DAPI and antipneumolysin antibody. Bar, 10 μm. Arrows indicate pneumolysin around or in the bacterium. **l** FIP200 KO MEFs infected with indicated pneumococcal strains for 1 h in the presence of 50 nM LysoTracker were stained with DAPI, and percentages of LysoTracker-positive PcV-containing cells were quantified. **m** Indicated MEFs were infected with pneumococci for 1 h and intracellular survivability of bacteria was determined by colony forming units (cfu); $n = 3$. Data are expressed as mean ± SEM; *$P < 0.01$, **$P < 0.05$, N.S., not significant. Uncropped blots for (**j**) can be found in Supplementary Fig. 10.

PcAV proceed simultaneously or sequentially, we attempted to construct PcLV-deficient MEFs. The C-terminal WD domain (WD; structural domain composed of ~40 amino acids, often terminated by a tryptophan–aspartic acid) of Atg16L1 plays a crucial role in the LAPosome-like autophagic bodies induced by entosis and certain bacteria[12,15,18]. We confirmed that canonical autophagy and flux were normal in Atg16L1 KO MEFs complemented with Atg16 FL (Full Length) or Atg16L1 lacking the WD (ΔWD) (Fig. 2a and Supplementary Fig. 2A–D). Next, we infected cells stably expressing GFP-LC3 with *S. pneumoniae* for 1 h. PcLV formation was completely abolished in Atg16L1 KO MEFs complemented with Atg16L1 ΔWD (Atg16L1 KO/ΔWD MEFs) (Supplementary Fig. 2B and D, left), suggesting that the Atg16L1 WD also plays a pivotal role also PcLV formation. We then investigated the intracellular localization of GFP-Atg16L1 ΔWD and WD at 1 h p.i. Only GFP-Atg16L1 FL was recruited to PcLVs (Supplementary Fig. 2E), indicating that both Atg5 binding by the N terminus and poly-Ub binding by the C terminus are required for appropriate localization. Next, we investigated PcAV formation using Atg16L1 KO/ΔWD MEFs. At 2 h p.i. in these PcLV-deficient MEFs, PcAV formation was robustly suppressed in an Atg16L1 WD-dependent manner (Fig. 2c, d, right). In light of a report that xenophagy is normal in Atg16L1 KO/ΔWD MEFs[16], these results clearly suggest that PcLV and PcAV proceed sequentially, rather than simultaneously (Fig. 2n).

Next, we investigated the localization of p62 in Atg16L1 KO MEFs complemented with Atg16L1 FL or ΔWD at 1 h p.i. p62-3Myc recruitment to intracellular bacteria was dramatically reduced in Atg16L1 KO/ΔWD MEFs (Fig. 2e and Supplementary Fig. 2F), indicating that the recruitment of p62 and Atg16L1 WD to PcLVs is interdependent. Importantly, poly-Ub deposition on PcVs was also robustly reduced in Atg16L1 KO/ΔWD MEFs (Fig. 2f and Supplementary Fig. 2G), although endosomal membrane rupture occurred normally in Atg16L1 KO/ΔWD MEFs at 1 h p.i. (Fig. 2g). LC3 deposition via Atg16L1 WD also participates in membrane repair in a $Ca^{2+}$-dependent manner[19]. However, the calcium chelator O,O′-bis(2-aminophenyl)ethyleneglycol-N,N,N′,N′-tetraacetic acid, tetraacetoxymethyl ester (BAPTA-AM) had no reductive effect on PcLV formation (Fig. 2h). Together, these results reveal the Atg16L1 WD-dependency of p62 and poly-Ub recruitment during PcLV formation.

Next, we investigated domain analysis of Atg16L1 WD domain in PcLV formation. The alpha domain of Atg16L1 (Fig. 2a) plays a role in PI3P-independent noncanonical autophagy induced by monensin[20]. Accordingly, we investigated PcLV formation using Atg16L1 KO MEFs complemented with the α or γ form of Atg16L1 FL. Intriguingly, PcLV formation in MEFs complemented with the α form was normal (Fig. 2i), suggesting that PcLVs are distinct from monensin-triggered LAPs. A coding polymorphism of human Atg16L1 (rs2241880; T300A) increases the risk of Crohn's disease; the T300A mutation (corresponding to T316A in mouse Atg16L1γ, hereafter referred to as T300A, Fig. 2a) in the WD region of Atg16L1 decreases the autophagic clearance of intracellular pathogens such as *Shigella*[21,22], but not *Salmonella*. Hence, we compared PcLV formation between Atg16L1 KO MEFs complemented with Atg16L1 WT or T300A where Atg14L had been knocked down to suppress canonical autophagy. We found that PcLV formation was reduced in T300A-complemented cells (Fig. 2j, and Supplementary Fig. 3A and B). Notably, subsequent PcAV formation and bactericidal activity at 2 h p.i. was also reduced in T300A-complemented cells (Fig. 2k, l), providing further confirmation that PcLV and PcAV were induced sequentially rather than simultaneously. Nod2 polymorphism is also a risk factor for Crohn's disease, and Nod2 and Atg16L1 work cooperatively in bactericidal autophagy[23,24]. Consistent with this, we found that PcLV formation was also reduced in Nod2-knockdown cells (Fig. 2m, and Supplementary Fig. 3C and D), implying a correlation between Crohn's disease and the severity of pneumococcal infections such as pneumonia and IPD[25,26]. Together, these results support the notion that PcLV and PcAV are induced sequentially during *S. pneumoniae* infection (Fig. 2n).

PcLV formation peaked at 1 h p.i. and robustly decreased at 2 h p.i. (Fig. 1b), leading us to hypothesize the existence of uncharacterized autophagic intermediates between PcLVs and PcAVs. It is also known that, during *Listeria* infection, the LC3 signal temporarily disappears after LAPosome formation, but then subsequently reappears around intracellular bacteria contemporaneous with xenophagy[11]. To verify our hypothesis, we first examined the kinetics of p62 and NDP52 recruitment to PcLVs. p62 and NDP52 were recruited to bacteria at similar levels at 1 h p.i., but NDP52 recruitment decreased by 2 h p.i. (34–14%), whereas p62 recruitment was unchanged (37–36%) (Fig. 3a), suggesting that NDP52 plays only a temporary role in PcLV formation, whereas p62 plays a role in formation of both PcLVs and PcAVs. This result is consistent with our previous report showing that p62 is frequently localized to PcAVs, whereas NDP52 localized in this manner very rarely[14]. The kinetics of NDP52-positive bacteria were reminiscent of the kinetics of the temporary appearance and disappearance of PcLVs, leading us to hypothesize that the disappearance of LC3 and NDP52 from

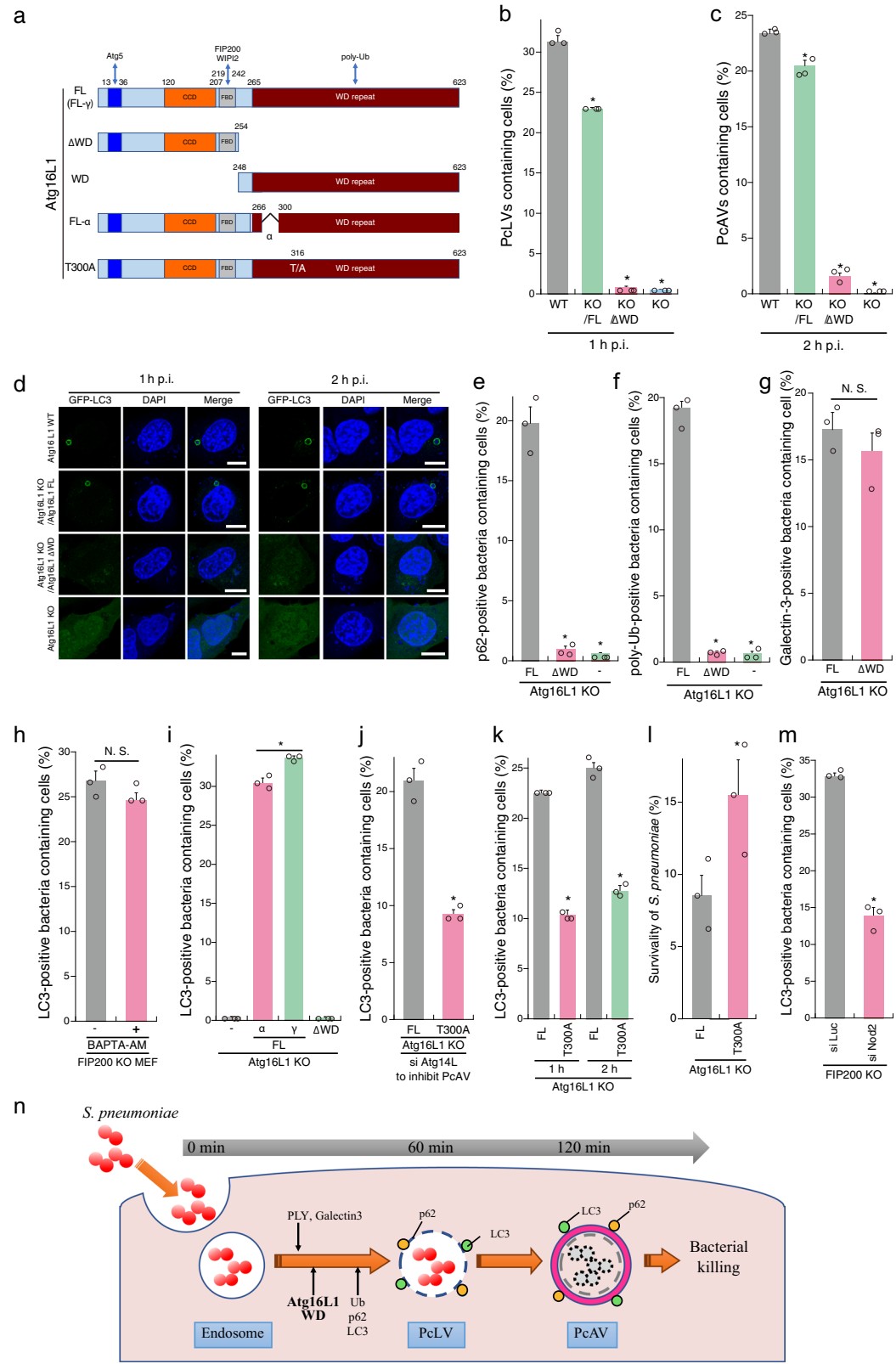

PcLVs is necessary for PcAV formation (Fig. 3j). Accordingly, we decided to designate these NDP52- and LC3-dissociated but p62-positive PcLVs as NDP52-delocalized PcLVs. We next examined the effect of Atg14L knockdown on PcLVs disappearance, i.e., NDP52-delocalized PcLVs appearance, in FIP200 KO MEFs infected with *S. pneumoniae* for 1, 2, or 4 h. The reduction in the abundance of PcLV-containing cells at 2 and 4 h p.i. was dramatically suppressed by Atg14L knockdown, suggesting that Atg14L is required for delocalization of LC3 from PcLVs (i.e., NDP52-delocalized PcLVs formation) (Fig. 3b, d). Similarly,

**Fig. 2 Atg16L1 WD domain is indispensable for PcLV formation and subsequent PcAV formation. a** Diagram of Atg16L1-derivatives used in this study. **b**, **c** Complemented Atg16L1 KO MEFs/GFP-LC3 infected with pneumococci for 1 or 2 h were stained with DAPI, and percentages of PcLV-containing cells were quantified. **d** Representative epifluorescence images in (**b**) and (**c**). **e**, **f** Complemented Atg16L1 KO MEFs/GFP-LC3 transiently expressing p62-3Myc were infected with pneumococci for 1 h and stained with DAPI and anti-Myc or -poly-Ub antibodies, and percentages of each marker-positive bacteria containing cells were quantified. **g** Complemented Atg16L1 KO MEFs/GFP-LC3 were infected with pneumococci for 1 h and stained with DAPI and anti-Galectin3 antibody and percentages of Galectin3-positive bacteria containing cells were quantified. **h** FIP200 KO MEFs/GFP-LC3 infected with pneumococci for 1 h with or without BAPTA-AM were stained with DAPI, and percentages of PcLV-containing cells were quantified. **i** Complemented Atg16L1 KO MEFs/GFP-LC3 were infected with pneumococci for 1 h and stained with DAPI and percentages of PcLV-containing cells were quantified. **j** Complemented Atg16L1 KO MEFs /GFP-LC3 treated with siAtg14L were infected with pneumococci for 1 h and stained with DAPI, and percentages of PcLV-containing cells were quantified. **k** Complemented Atg16L1 KO MEFs /GFP-LC3 infected with pneumococci for 1 or 2 h were stained with DAPI, and percentages of LC3-positive bacteria containing cells were quantified. **l** Complemented Atg16L1 KO MEFs were infected with pneumococci for 2 h and intracellular survivability of bacteria was determined by colony forming units (cfu); $n = 3$. **m** FIP200 KO MEFs/GFP-LC3 treated with indicated siRNA were infected with pneumococci for 1 h and stained with DAPI, and percentages of PcLV-containing cells were quantified. **n** Schematic diagram of PcLV- and subsequent PcAV formation. Data are expressed as mean ± SEM.; *$P < 0.01$, N.S., not significant. Bar, 10 μm.

NDP52 delocalization from PcLVs at 2 h p.i. was clearly suppressed by Atg14L knockdown (Fig. 3c, d), suggesting that the kinetics of LC3 and NDP52 delocalization from PcLVs are similar. We next estimated NDP52-positive/LC3-positive vacuoles. At 1 h p.i., the ratio of NDP52$^+$/LC3$^+$ vacuoles was 75%, but it was reduced to 45% at 2 h p.i., indicating that NDP52 delocalization occurred more rapidly than LC3 delocalization (Fig. 3e). We next examined whether Atg14L-dependent NDP52-delocalized PcLVs transition was also observed in FIP200 WT MEFs. The population of LC3-positive PcV-containing cells at 1 h p.i. was not reduced by Atg14L knockdown in FIP200 WT MEFs (Fig. 3f and Supplementary Fig. 4A), suggesting that PcLV formation ordinarily occurs even in WT MEFs at 1 h p.i. Notably, LC3 delocalization from PcLVs at 2 h p.i. was not suppressed by Atg14L knockdown in FIP200 WT MEFs (Fig. 3f and Supplementary Fig. 4A). As Atg14L knockdown can suppress subsequent PcAVs, there was a possibility that we underestimated the abundance of PcLVs. Thus, we quantified NDP52$^+$/LC3$^+$ vacuoles in siAtg14L-treated FIP200 WT MEFs, and we concluded that NDP52-delocalized PcLVs transition (i.e., delocalization of NDP52 from PcLVs) was not suppressed by Atg14L knockdown in FIP200 WT MEFs (Fig. 3g). Together, these results indicate that when either Atg14L or FIP200 is present (i.e., in FIP200 KO MEFs or in siAtg14L-treated WT MEFs), the NDP52-delocalized PcLVs transition can proceed without hindrance (i.e., NDP52 stayed associated with p62 only when both of FIP200 and Atg14L were absent) (Fig. 3j). Notably, as NDP52-delocalized PcLVs transition proceeded even in siAtg14L-treated FIP200 WT MEFs, we presumed that LC3 and NDP52 delocalization from PcLVs was not due to the elimination of PcLVs by canonical autophagy, which requires Atg14L.

Next, we investigated whether the delocalization of LC3 from PcLVs was suppressed by Beclin1 depletion, because Atg14L and Beclin1 are both essential components in the PI3KC3 complex for PI3P production. The results suggested that PI3P production via PI3KC3 is required for NDP52-delocalized PcLVs transition (Fig. 3h and Supplementary Fig. 4B and D). Myo6–NDP52 interaction is involved in tethering NDP52-positive vacuoles with lysosomes[27]. Upon knockdown of myosin VI, no suppressive effect on NDP52-delocalized PcLVs transition was observed (Fig. 3i and Supplementary Fig. 4C and E), suggesting that LC3–NDP52–Myo6-dependent endosome–lysosome fusion is not involved in NDP52 delocalization from PcLVs. Notably, PcLV formation was not affected by Beclin1 or Myo6 depletion. Together, these results support the notion that PcLVs, which are NDP52 and LC3 positive, and NDP52-delocalized PcLVs, which are NDP52 negative and p62 positive, proceed sequentially during *S. pneumoniae* infection (Fig. 3j).

We showed above that multiple autophagy processes involving PcLV, NDP52-delocalized PcLV, and PcAV sequentially proceed during *S. pneumoniae* infection (Figs. 2n and 3j), leading us to determine a detailed time course of these hierarchical autophagy processes in *S. pneumoniae*-infected cells. At first, we investigated the kinetics of the temporary appearance of LC3- or NDP52-positive PcLVs in FIP200 KO MEFs. Notably, LC3-positive PcLVs was not observed until at 30 min p.i. However, it acutely appeared at 60 min p.i. and gradually disappeared by 150 min after infection (Fig. 4a). Similarly, NDP52-positive PcLVs acutely increased at 60 min p.i. and gradually disappeared by 180 min after infection (Supplementary Fig. 5). Next, we observed the kinetics of NDP52- or LC3-positive pneumococci-containing vacuoles in FIP200 WT MEFs. Consistent with Fig. 4a, NDP52-positive pneumococci-containing vacuoles (i.e., PcLVs) acutely appeared at 60 min p.i. and gradually disappeared by 150 min p.i. (Fig. 4b and Supplementary Fig. 6A). LC3-positive pneumococci-containing vacuoles also rapidly increased at 60 min p.i. However, it transiently decreased at 90 min p.i. and again increased by 150 min p.i. (Fig. 4b), indicating that NDP52-delocalized PcLVs appeared at ~90 min p.i. (Supplementary Fig. 6B). We designated ([LC3-positive pneumococci-containing vacuoles containing cells] − [NDP52-positive pneumococci-containing vacuoles containing cells]) as a putative population of PcAV-containing cells (Supplementary Fig. 6A), showing that PcAV was gradually induced after 90 min of infection and proceeded until 180 min after infection (Supplementary Fig. 6B). Next, we examined the kinetics of p62- or LC3-positive PcLVs in FIP200 KO MEFs. Consistent with Fig. 4a, LC3-positive PcLVs acutely appeared at 60 min p.i. and gradually decreased (Fig. 4c). However, in accordance with Fig. 3a, the abundance of p62-positive pneumococci containing-vacuoles containing cells was not changed until 120 min p.i. (Fig. 4c), indicating that NDP52-delocalized PcLV was induced after 90 min of infection and continued until 120 min p.i. (Supplementary Fig. 6B) in FIP200 KO MEFs. We designated ([p62-positive pneumococci-containing vacuoles containing cells] − [LC3-positive pneumococci-containing vacuoles containing cells]) as a putative population of NDP52-delocalized PcLV-containing cells (Supplementary Fig. 6A), showing that NDP52-delocalized PcLV is induced at interphase between PcLV and PcAV at 90 min p.i. (Supplementary Fig. 6A). Finally, we examined the kinetics of NDP52- or p62-positive pneumococci-containing vacuoles in FIP200 KO MEFs. Consistent with Fig. 4b and Supplementary Fig. 5, NDP52-positive pneumococci containing-vacuoles containing cells rapidly increased at 60 min p.i. and gradually disappeared by 180 min p.i., however, the abundance of p62-positive pneumococci containing-vacuoles containing cells was

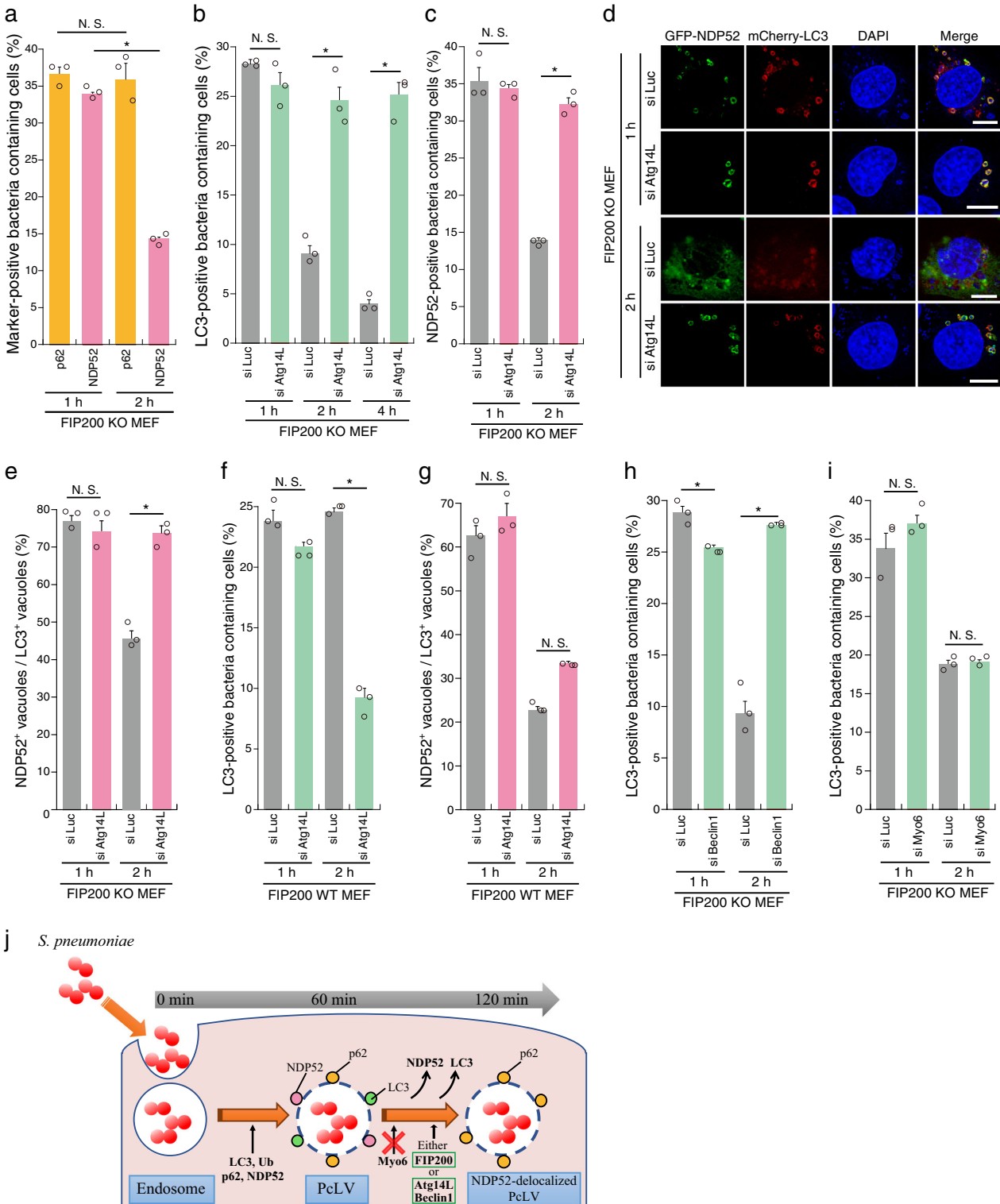

**Fig. 3 Atg14L, Beclin1, and FIP200 are involved in NDP52-delocalized PcLVs transition. a** FIP200 KO MEFs/mCherry-LC3 stably expressing GFP-p62 or GFP-NDP52 infected with pneumococci for 1 or 2 h were stained with DAPI, and percentages of each marker-positive bacteria containing cells were quantified. **b** FIP200 KO MEFs/GFP-LC3 treated with indicated siRNA were infected with pneumococci for 1, 2, or 4 h and stained with DAPI, and percentages of LC3-positive bacteria containing cells were quantified. **c–e** FIP200 KO MEFs/mCherry-LC3/GFP-NDP52 treated with indicated siRNA were infected with pneumococci for 1 or 2 h and stained with DAPI, and percentages of NDP52-positive bacteria containing cells and the populations of GFP-NDP52$^+$/mCherry-LC3$^+$ vacuoles were quantified. **d** Representative epifluorescence images in (**c**). **f, g** FIP200 WT MEFs/mCherry-LC3 treated with indicated siRNA were infected with pneumococci for 1 or 2 h and stained with DAPI, and percentages of PcLV-containing cells were quantified. **h, i** FIP200 KO MEFs/GFP-LC3/GFP-NDP52 treated with indicated siRNA were infected with pneumococci for 1 or 2 h and stained with DAPI, and percentages of LC3-positive bacteria containing cells and the populations of GFP-NDP52$^+$/mCherry-LC3$^+$ vacuoles were quantified. **j** Schematic diagram of PcLV- and subsequent NDP52-delocalized PcLV. Data are expressed as mean ± SEM.; *$P < 0.01$, N.S., not significant. Bar, 10 μm.

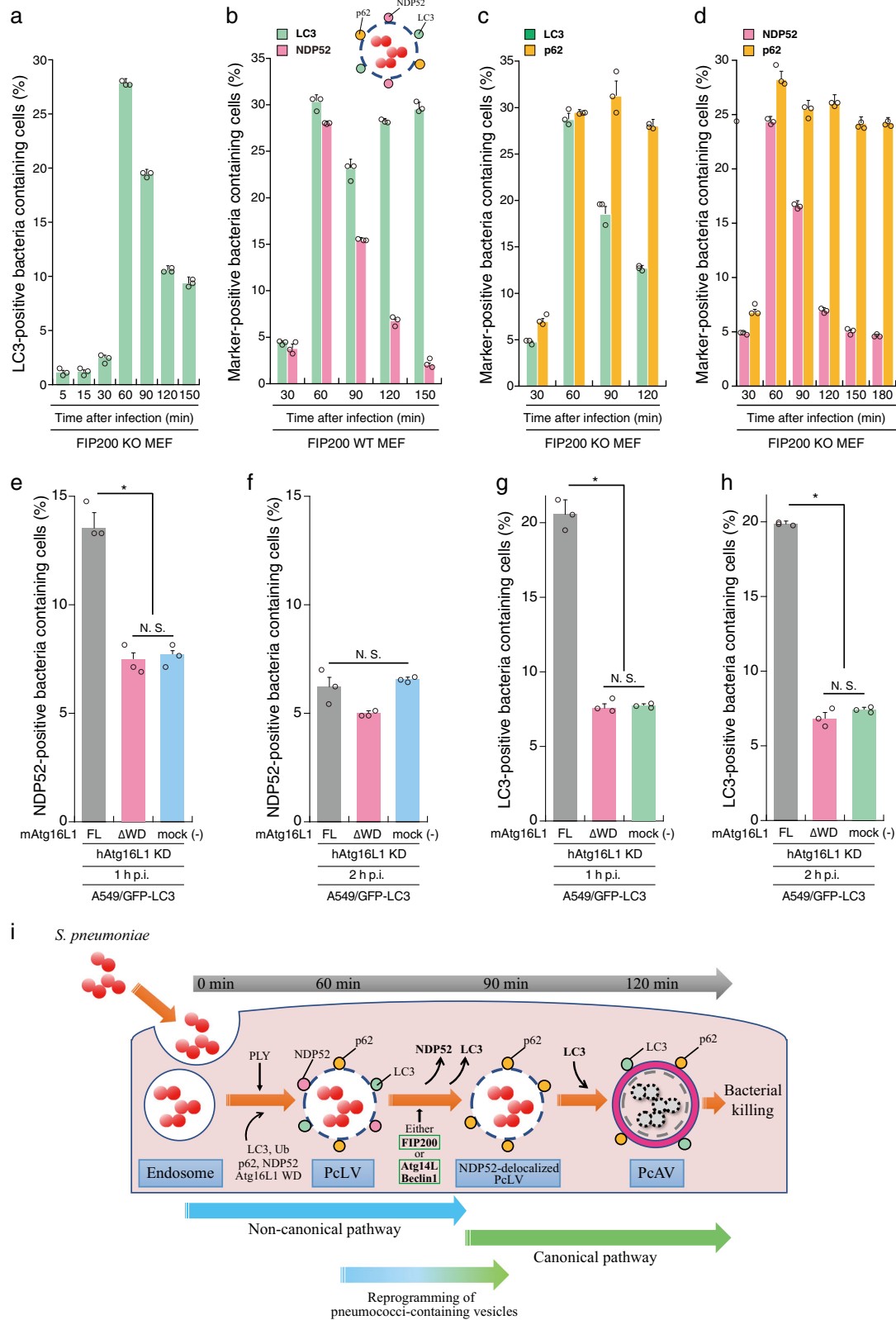

almost unchanged until 180 min p.i. (Fig. 4d). Together, these results supported the notion that multiple hierarchical autophagy processes, including PcLV, NDP52-delocalized PcLV, and eventually PcAV, sequentially proceed during *S. pneumoniae* infection (Fig. 4i and Supplementary Fig. 6B).

Next, we confirm whether our findings observed in MEF cells can be replicated in human pulmonary epithelial cells (A549 cells). We investigated the kinetics of the temporary induction of PcLVs in siLuc- or siFIP200-treated A549 cells. Consistent with the results in MEFs, FIP200-independent PcLV was transiently

**Fig. 4 Hierarchical autophagy processes during time course of pneumococcal infection. a** FIP200 KO MEFs/GFP-LC3 infected with pneumococci for indicated periods were stained with DAPI, and percentages of GFP-LC3 positive bacteria containing cells were quantified. **b** FIP200 WT MEFs/mCherry-LC3/GFP-NDP52 infected with pneumococci for indicated periods were stained with DAPI, and percentages of each marker-positive bacteria containing cells were quantified. **c** FIP200 KO MEFs/GFP-LC3 infected with pneumococci for indicated periods were stained with anti-p62 antibody and DAPI, and percentages of each marker-positive bacteria containing cells were quantified. **d** FIP200 KO MEFs/GFP-NDP52 infected with pneumococci for indicated periods were stained with anti-p62 antibody and DAPI, and percentages of each marker-positive bacteria containing cells were quantified. **e–h** A549 cells/ GFP-LC3/mouse Atg16L1 FL or ΔWD were treated with siRNA for human Atg16L1. After 2 days, the cells were infected with pneumococci for 1 or 2 h, and stained with anti-NDP52 antibody and DAPI, and percentages of each marker-positive bacteria containing cells were quantified. **i** Schematic diagram of pneumococci-containing endosomes-remodeling to PcAVs through PcLV and NDP52-delocalized PcLV. Data are expressed as mean ± SEM.; *$P < 0.01$, N.S., not significant.

induced in A549 cells (Supplementary Fig. 7A–B and F–G). Notably, at 3 and 4 h p.i., LC3 recruitment remained high in siLuc-treated cells. However, it was decreased in siFIP200-treated cells, suggesting that subsequent PcAV is induced in siLuc-treated cells at 3 and 4 h p.i. (Supplementary Fig. 7C). In this experimental setting (i.e., siFIP200 knockdown for 3 days in A549 cells), PcLVs appearance was little later than that in MEFs (between 1 and 2 h). Treatment with antioxidants, such as apocynin or NAC, had no effect on PcLV formation in siFIP200-treated A549 cells, suggesting that PcLV in A549 cells is also ROS independent (Supplementary Fig. 7D and E). Finally, we performed siRNA rescue experiments using A549 cells stably expressing mouse Atg16L1 (Supplementary Fig. 7H). When these A549 cells were infected with *S. pneumoniae* under Atg16L1-knockdowned conditions for 1 or 2 h, NDP52-positive PcLVs appeared at 1 h p.i. and disappeared at 2 h p.i. (Fig. 4e, f). By contrast, in siRNA-resistant Atg16L1 WD rescued or mock (no rescued) A549 cells, the formation of NDP52-positive PcLV was suppressed (Fig. 4e, f). Next, when LC3 localization was quantified, LC3-positive PcLVs appeared at 1 h p.i. in siRNA-resistant Atg16L1 FL-rescued A549 cells. However, it was robustly decreased in siRNA-resistant Atg16L1 WD rescued or mock (no rescued) A549 cells (Fig. 4g), which was quite similar to the result in Atg16L1-complemented and Atg16L1 KO MEFs (Fig. 2b). At 2 h p.i., where NDP52 had already disappeared from PcLVs (Fig. 4f), PcAV was apparently observed in siRNA-resistant Atg16L1 FL-rescued A549 cells, but it was robustly suppressed in siRNA-resistant Atg16L1 WD rescued or mock A549 cells (Fig. 4h). These findings were also similar to those in Atg16L1-complemented and Atg16L1 KO MEFs (Fig. 2c). These results clearly show that multiple hierarchical autophagy processes also occurred in human pulmonary epithelial cells.

**Poly-Ub, p62, NDP52, and Atg16L1 WD are interdependently involved in PcLV formation**. Although we previously reported that NDP52 was rarely recruited to PcAVs at 2 h p.i.[14], NDP52 was frequently recruited to PcLVs at 1 h p.i. at a level comparable with p62 (Fig. 3a); however, their contribution to PcLV formation remains unclear. Hence, we investigated the interdependence of poly-Ub, p62, and NDP52 in PcLV formation. Treatment with PYR-41, a ubiquitin-activating enzyme (E1) inhibitor, led to a reduction in the number of LC3-, poly-Ub-, p62-, and NDP52-positive bacteria at 1 h p.i. in FIP200 KO MEFs (Fig. 5a, b). Next, we performed a knockdown experiment to clarify the role of p62 and NDP52 in PcLV formation. Upon depletion of p62, NDP52, or TBK1 (activator kinase of these adapters), PcLV formation was dramatically reduced, suggesting that both p62 and NDP52 are necessary for PcLV formation (Fig. 5c and Supplementary Fig. 8A–C). Consistent with this, PcLV formation was reduced in Atg14L-knockdown p62 KO MEFs (Supplementary Fig. 8E and F). Intriguingly, poly-Ub deposition on PcVs was also reduced in p62-, NDP52-, or TBK1-knockdown FIP200 KO MEFs, suggesting that cargo receptors promote poly-Ub deposition on

PcVs, and vice versa (Fig. 5d). Next, we assessed the interdependence of p62 and NDP52 recruitment to intracellular *S. pneumoniae* in FIP200 KO MEFs. Upon depletion of p62 or TBK1, we observed a robust decrease in NDP52 recruitment to intracellular bacteria (Fig. 5e). Similarly, upon depletion of NDP52 or TBK1, p62 recruitment to bacteria was also reduced (Fig. 5f), suggesting that the recruitment of p62 and NDP52 to intracellular pneumococci is interdependent. Galectin-8 is a target of NDP52 in xenophagy[5]. Upon depletion of Galectin-8, NDP52 recruitment was also abolished, but the inhibition of PcLV formation was partial (Fig. 5g and Supplementary Fig. 8D). Intracellular pneumococci can evade into the cytosol[28]. We then investigated whether LC3, p62, and poly-Ub deposition is targeted on the endosomal membrane-engulfing bacteria or on the cytosolic bacterial surface. In PcLV and NDP52-delocalized PcLV, these signals were clearly decorated on the membrane-engulfing bacteria in a ring shape (Fig. 5h). Similarly, when the phosphoserine-decorated membrane was visualized by GFP-Lact-C2, and most of the intracellular pneumococci were still in the membranous compartment even at the NDP52-delocalized PcLV stage (Fig. 5i). This finding suggests that most of the intracellular pneumococci remained in the damaged vacuoles during sequential PcLV and NDP52-delocalized PcLV.

Together, these results clearly showed that poly-Ub, p62, NDP52, and Atg16L WD (Fig. 2e, f) are mutually dependent for their recruitment to intracellular pneumococci, and that all of these proteins are required for PcLV formation (Fig. 5j).

Hence, we sought to assess the interdependence of p62, NDP52, and Atg16L1 in PcLV formation. To this end, we investigated whether p62 or NDP52 can interact with Atg16L1 through WD. When GFP-Atg16L1 was immunoprecipitated in 293T cells transiently expressing GFP-Atg16L1 and p62-3Myc or NDP52-3Myc, p62 was coprecipitated with GFP-Atg16L1, but NDP52 was not (Fig. 6a). p62 and Atg16L were clearly colocalized with PcLVs (Fig. 6b). Hence, we performed a domain analysis of Atg16L1 by immunoprecipitation (IP). The results revealed that the C-terminal portion of Atg16L1 (WD domain) is responsible for p62 binding (Fig. 6c). Structurally, p62 contains six functional domains: the PB1 (Phox and Bem1p), ZZ (ZZ-type zinc finger), TBS (TRAF6-biding sequence), LIR (LC3-interacting region), KIR (Keap1-binding region), and UBA (ubiquitin-associated) domains (Fig. 6d). We prepared a series of truncated versions of p62-3Myc, including K7A/D69A (dimerization defective), ΔLIR, ΔUBA, or ΔLIR/ΔUBA (Fig. 6d), and investigated their ability to interact with Atg16L1 WD by IP experiment. We found that the p62–Atg16L1 WD interaction was dramatically reduced in the mutant lacking UBA (Fig. 6e). In addition, we prepared other p62-3Myc variants, including ΔZZ (RIP1-binding defective), ΔTBS, and ΔKIR (Fig. 6d), and tested their ability to interact with Atg16L1 WD. We also found that the p62–Atg16L1 WD interaction was dramatically reduced in the ΔTBS mutant (Fig. 6f). These results suggest that the poly-Ub–p62 and TRAF6–p62 interactions are important to the p62–Atg16L1 WD interaction.

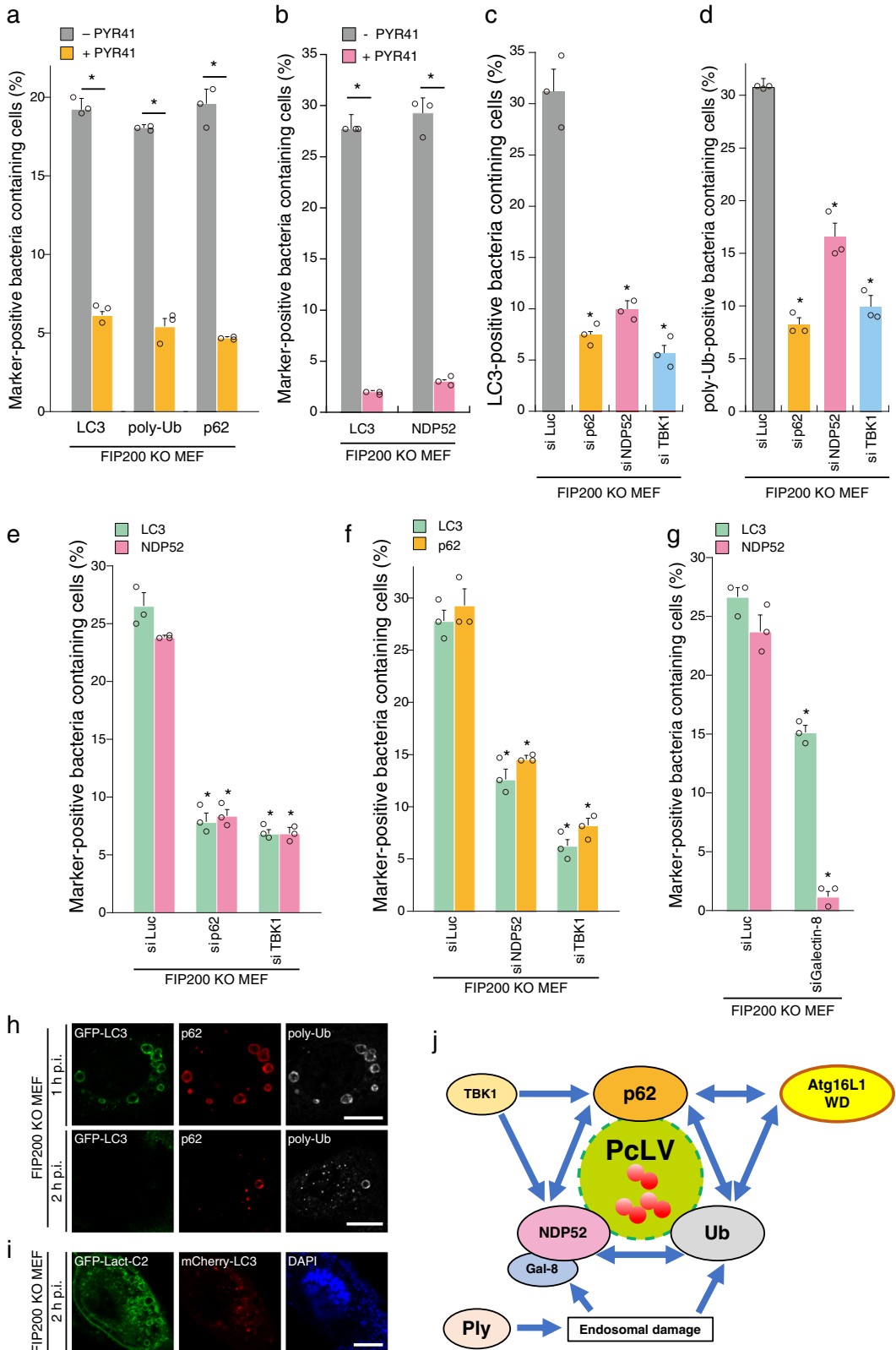

To explore the involvement of TRAF6-mediated poly-Ub chain formation in the p62–Atg16L1 interaction, we performed IP in the presence or absence of FLAG-TRAF6. Endogenous p62 was coimmunoprecipitated with GFP-Atg16L1 in the presence of FLAG-TRAF6, and FLAG-TRAF6 was also coimmunoprecipitated with GFP-Atg16L1 (Fig. 6g). Importantly, binding of K63-linked poly-Ub chain (K63Ub) to Atg16L1 was dramatically elevated in the presence of FLAG-TRAF6, but Atg16L1 was not ubiquitinated (Fig. 6g). These results suggest that TRAF6-generated K63Ub on an unknown target may be involved in the interaction between the UBA of p62 and the WD of Atg16L1 (Fig. 6j). Next, we performed domain analysis of the

**Fig. 5 Poly-Ub, p62, and NDP52 regulate PcLV formation interdependently. a, b** FIP200 KO MEFs/GFP-LC3 or FIP200 KO MEFs/mCherry-LC3/GFP-NDP52 infected with WT or Δ*ply* pneumococci for 1 h with or without PYR-41 were stained with DAPI and antibodies against poly-Ub or p62, and percentages of each marker-positive bacteria containing cells were quantified. **c, d** FIP200 KO MEFs/mCherry-LC3 treated with indicated siRNA were infected with pneumococci for 1 h and stained with DAPI and anti-poly-Ub antibody, and percentages of LC3- or poly-Ub-positive bacteria containing cells were quantified. **e, f** FIP200 KO MEFs/mCherry-LC3 stably expressing GFP-p62 or GFP-NDP52 treated with indicated siRNA were infected with pneumococci for 1 h and stained with DAPI, and percentages of each marker-positive bacteria containing cells were quantified. **g** FIP200 KO MEFs/mCherry-LC3/GFP-NDP52 treated with indicated siRNA were infected with pneumococci for 1 h and stained with DAPI, and percentages of each marker-positive bacteria containing cells were quantified. **h, i** FIP200 KO MEFs/GFP-LC3 or FIP200 KO MEFs/mCherry-LC3/GFP-Lact-C2 infected with WT pneumococci for 1 or 2 h were stained with DAPI and antibodies against p62 or poly-Ub. **j** Autophagy machinery protein dynamics in PcLV formation. Data are expressed as mean ± SEM.; *$P < 0.01$.

Atg16L1–TRAF6 interaction by IP, and we found that the T300A substitution did not negatively affect the Atg16L1–TRAF6 and TRAF6-enhanced Atg16L1–p62 interactions (Supplementary Fig. 9A). Notably, TRAF6 can bind to the Atg16L1 N-terminal ΔWD domain (Fig. 6j and Supplementary Fig. 9B–C), where it is involved in Atg5, FIP200, or WIPI-2 binding. Atg16L1 harbors a conserved TRAF6-interacting motif in the central region (Supplementary Fig. 9B), but the this putative TRAF6-binding region of Atg16L1 was not involved in TRAF6 binding (Supplementary Fig. 9C). Next, we investigated whether GFP-TRAF6 was recruited to PcLVs, and found that GFP-TRAF6 was clearly confined to PcLVs (Fig. 6h). Finally, PcLV formation was robustly decreased by TRAF6 knockdown in FIP200 KO MEFs (Fig. 6i and Supplementary Fig. S9D).

## Discussion

In this study, we identified LC3- and NDP52-delocalized LAPosome-like vacuoles and found evidence that multiple autophagic processes involving PcLV, NDP52-delocalized PcLV, and PcAV sequentially target intracellular *S. pneumoniae* (Fig. 4i). At early stage of infection, intracellular pneumococci evade the bactericidal endocytic pathway by triggering PcLV, and take refuge in a transient niche; however, subsequent PcAVs eventually destroy this survival niche. Furthermore, we demonstrated that several components of autophagic machinery, including poly-ubiquitin, p62, NDP52, and Atg16L1, are interdependently recruited to damaged PcVs during LAP-like autophagic processes (Fig. 5j), and that TRAF6-mediated poly-Ub formation promotes the p62–Atg16L1 interaction (Fig. 6j) and subsequent PcLV formation. Furthermore, we suggested that Crohn's disease might be correlated with pneumococcal diseases (Fig. 2j–m).

Multiple autophagic pathways and distinct autophagic machineries are employed during distinct bacterial infections[2]. Indeed, p62 and NDP52 are interdependently recruited to intracellular *Shigella* during xenophagy, whereas poly-Ub, p62, and NDP52 are independently recruited to intracellular *Listeria* during this process[29]. Moreover, p62 and NDP52 are independently recruited to intracellular *Salmonella* during xenophagy[30]. Thus, the interdependence of p62 and NDP52 is distinct in each bacterial species. In *S. pneumoniae* infection, p62 is required for both PcLV and PcAV formation, but NDP52 is involved only in PcLV formation. In addition, we found that poly-Ub, p62, and NDP52 are mutually dependent for their recruitment to intracellular pneumococci, and all are required for PcLV formation. Importantly, we obtained evidence of LC3- and NDP52-delocalized but p62-positive PcLVs during the transition from PcLVs to PcAVs. However, the roles of Atg14L, Beclin1, and FIP200 during the NDP52-delocalized PcLVs transition, as well as the events that occur during the disappearance of LC3 and NDP52, remain to be elucidated.

In LAP and LAP-like autophagy, the involvement of cargo receptors in each experimental setting is distinct. Poly-Ub and p62 are recruited to vacuoles induced by LLOMe, a lysosome-specific inducer of membrane damage[31]. In AMDE-1-induced LAP-like autophagy, p62 is recruited, but is not necessary[32]. By contrast, in monensin- or chloroquine-induced LAP-like autophagy, p62 and NDP52 are not recruited[33]. Similarly, *Listeria*-induced LAP in phagocytes requires LLO (Listeriolysin O)-induced membrane rupture, but not poly-Ub or p62[9]. In VacA-induced LAP-like autophagy process, p62 is not recruited[34]. In this study, we revealed that PcLVs are a previously unreported type of FIP200-independent LAP-like vesicles to which Galectin3, poly-Ub, p62, and NDP52 are recruited. In addition, PcLVs has unique features among LAP-like vesicles, including ROS-, TLR2-, PI3P-, Myo6-, and Atg16L1 α domain independence. Importantly, it remains to be elucidated whether a common molecule or biochemical event is involved in these LAP-like autophagy processes.

Atg16L1 T300A has no negative effect on *Salmonella*-induced LAP-like autophagy or xenophagy[16]. By contrast, Atg16L1 T300A is associated with reduced xenophagy against *Shigella*[21,22], and we found that PcLV and PcAV formation was reduced in Atg16L1 T300A-complemented cells. LAP-like autophagy and canonical autophagy proceed concurrently in *Salmonella* infection[15,16]. Hence, we speculated that Atg16L1 T300A would have a negative effect only on LAP or LAP-like processes, but not on canonical autophagy (including xenophagy) and that defects in LAP-like processes would be masked by canonical autophagy in *Salmonella* infection. It is most exciting to speculate that some pneumococcal virulence factor can manipulate or delay FIP200-dependent canonical autophagy during the early stage of infection in order to acquire a transient survival niche. The Atg16L1 T300A protein is susceptible to Caspase-3 and -7-mediated digestion in myeloid cells during apoptosis[22]. However, Atg16L1 T300A is not preferentially cleaved by *S. pneumoniae* infection (Supplementary Fig. 3E). Notably, pneumococcal infection tends to be more severe in Crohn's patients[25,26], and it is tempting to speculate that autophagy exerts a protective effect against severe or IPD.

In summary, several lines of evidence suggest that mammalian cells use several pathways to target bacteria for autophagy: canonical autophagy, Ub-dependent selective autophagy, and LAP-like autophagy, including the process involving PcLV. Indeed, *S. pneumoniae* can trigger autophagy via various pathways in multiple cell types. Our discovery gives insight into the mechanism by which the host defense system senses intracellular pulmonary bacterial pathogens. This knowledge may facilitate the development of therapeutic targets for the control of pneumococcal diseases.

## Methods

**Bacterial strains**. *S. pneumoniae* strain R6 (ATCC BAA-255) and strain ATCC6308 (ST8) were purchased from the American Type Culture Collection. Δ*ply* *S. pneumoniae* strain was constructed as described previously[14]. *S. pneumoniae* was grown as standing cultures in THY (Todd–Hewitt Broth (BD) plus 0.5% yeast extract (BD)) broth, or plated THY-agar plates, or Columbia agar plates with 5% sheep blood (BD) at 37 °C in a 5% $CO_2$ in air atmosphere. *Escherichia coli* strains MC1061, DH10B, or C43 (Cosmo Bio) were used for DNA cloning and

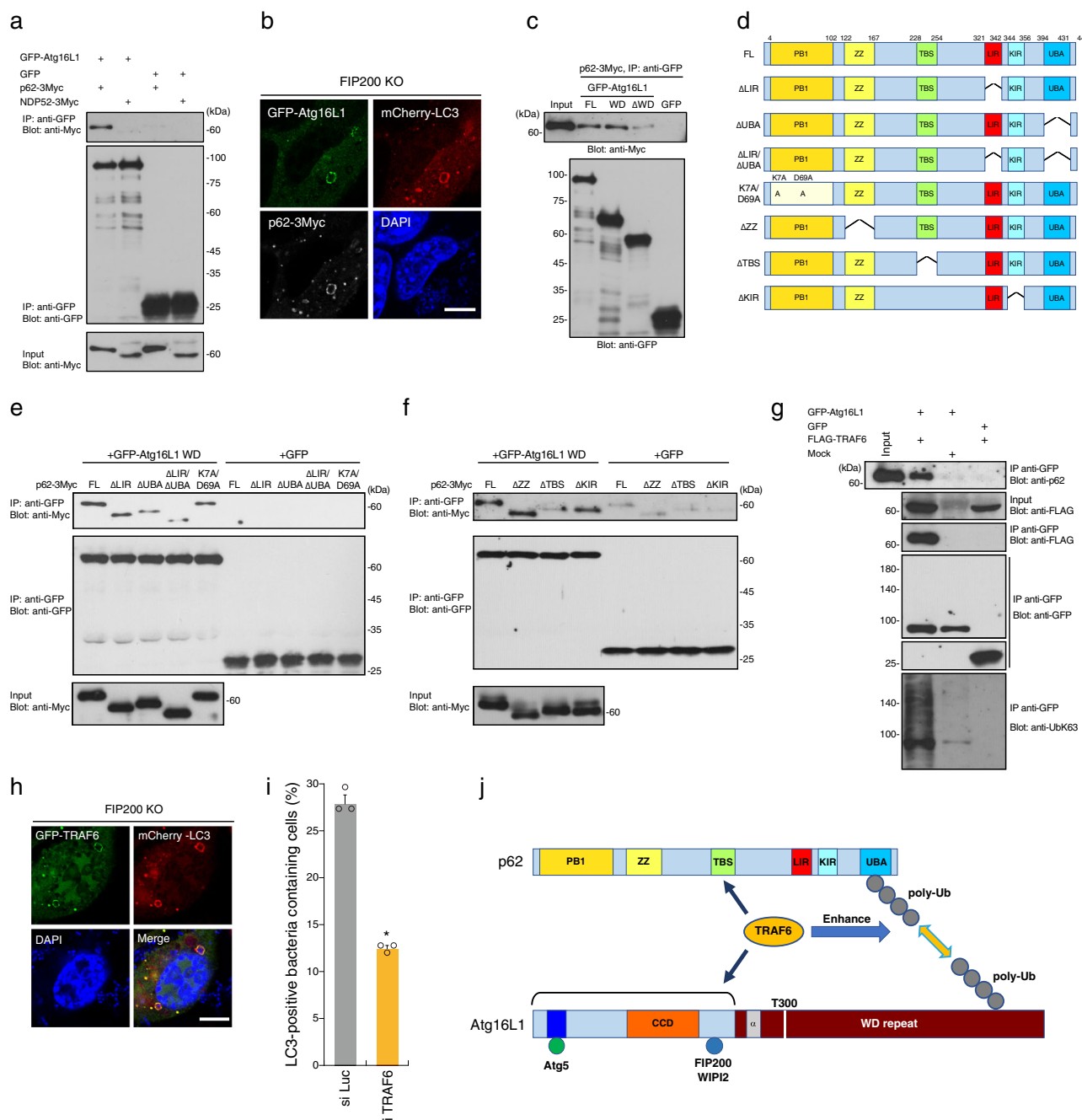

**Fig. 6 p62–Atg16L1 interaction mediated by poly-ubiquitin chain is involved in PcLV formation. a–c** Lysates of 293T cells expressing indicated proteins were pulled-down with GFP-nanobody, and the bound proteins were immunoblotted with indicated antibodies. **b** FIP200 KO MEFs/mCherry-LC3/p62-3Myc transiently expressing GFP-Atg16L1 were infected with pneumococci for 1 h, and stained with DAPI and anti-Myc antibody. **d** Schematic diagram of the domains of p62 and the p62 truncation constructs used in this study. **e–f** Lysates of 293T expressing indicated proteins were immunoprecipitated with GFP-nanobody, and the bound proteins were immunoblotted with indicated antibodies. **g** Lysates of 293T expressing indicated proteins were immunoprecipitated with GFP-Trap, and the bound proteins were immunoblotted with indicated antibodies. **h** FIP200 KO MEFs/mCherry-LC3 transiently expressing GFP-TRAF6 infected with pneumococci for 1 h were stained with DAPI. **i** FIP200 KO MEFs/GFP-LC3 treated with indicated siRNA were infected with pneumococci for 1 h and stained with DAPI, and percentages of PcLV-containing cells were quantified. **j** Schematic diagram of p62 and Atg16L1 WD interaction mediated by poly-Ub and TRAF6. Data are expressed as mean ± SEM.; *P < 0.01. Bar, 10 μm. Uncropped blots for **a**, **c**, **e**, **f**, and **g** can be found in Supplementary Fig. 11.

plasmid construction, and grown in LB broth or on LB agar plates, supplemented with 100 μg/ml ampicillin or 50 μg/ml kanamycin.

**Reagents and antibodies**. Anti-*S. pneumoniae* (SSI), anti-GFP (Cell Signaling), anti-Myc (9B11, Cell Signaling), anti-Flag (Wako), anti-Galectin-3 (SINO BIOLOGICAL), anti-Calcoco2 (Proteintech for WB), anti-NDP52 (Gene Tex (GTX115378) for IF), anti-K63 linked Ub (clone Apu3, EMD Millipore), anti-LC3, p62, RFP, Atg16L1, ubiquitin (MBL), and anti-actin (Santa Cruz Biotechnology, Inc.) were used as primary antibodies. An HRP-conjugated goat anti-rabbit or anti-

mouse antibodies (Jackson Laboratories) were used as secondary antibodies for immunoblotting. FITC- or TRITC-conjugated goat anti-rabbit or anti-mouse IgG antibodies (Sigma-Aldrich) were used as secondary antibodies for immunostaining. DAPI (4′,6-diamidino- 2-phenylindole, Sigma-Aldrich) was used for DNA staining. LysoTracker DND-99 was purchased from molecular probes. 10 μM rapamycin and 40 μM chloroquine (Selleck chemical), and 30 μM PYR-41 (UBPBio), and 10 mM 3-methyladenine (3-MA, Wako) were used as autophagy inducer or inhibitor. 300 μM Apocynin (Selleck), 10 mM GSH (Cayman Chemical), 2.5 mM NAC (Sigma-Aldrich) were used as antioxidative reagents. All other reagents were

purchased from Sigma-Aldrich. All antibodies were used at 1:100 for immuno-fluorescence staining and 1:1000 for western blotting.

**Plasmids.** All PCR was executed by Q5 High-Fidelity DNA Polymerase (New England BioLabs), and all RT-PCR was performed by SuperScript III One-Step RT-PCR System with Platinum Taq (Thermo Fisher Scientific). The vectors for GFP- or mCherry-rat LC3B, or human pIgR stable-expressing MEFs generation were constructed as described[14]. Vectors for GFP-Nedd4-1, GFP-Galectin-3[14], WIPI2-GFP, GFP-Rubicon[4], and p62 (FL, ΔLIR, ΔUBA, ΔLIR-ΔUBA, and K7A/D69A)-3Myc[6] expression vectors were constructed as described previously. pcDNA-p62-3Myc ΔZZ, ΔTBS, and ΔKIR were constructed by NEB builder (New England BioLabs) using pcDNA-p62-3Myc FL as a template and primers shown in Supplementary Table 1. For stable clone generation, obtained p62-3Myc ΔTBS cDNA was subcloned into pMXs-puro vector (Cosmo Bio). GFP-LC3 G120A was constructed with QuickChange (Agilent) using pEGFP-rat LC3B as a template and primers shown in Supplementary Table 1. For pcDNA3.1-NDP52-3Myc construction, human NDP52 cDNA amplified by PCR using pEGFP-NDP52 as a template and primers shown in Supplementary Table 1 was subcloned into pcDNA3.1-3Myc vector[4]. For the construction of NDP52-3Myc-stably expressing cells, obtained NDP52-3Myc cDNA was subcloned into pMXs-IRES-blast vector (Cosmo Bio). For construction of pEGFP-Ply variants, allele 1 *ply* gene of R6, or Allele 5 *ply* gene of ATCC6308 were subcloned into pEGFP-C1 vector (Clontech). For construction of pEGFP-mouse Atg16L1, mouse Atg16L1 cDNA purchase from Openbiosystems (IMAGE:6813377) was subcloned into pEGFP-C3 vector. For the construction of Atg16L1-stably expressing cells, nontagged mouse Atg16L1 cDNA was subcloned into pMXs-puro vector. For the construction of mouse Atg16L1 ΔWD or WD, each cDNA was amplified by PCR with primers shown in Supplementary Table 1. Human OPTN and UVRAG cDNA were amplified by RT-PCR using primer pairs shown in Supplementary Table 1 and total mRNA from Detroit 562 or HeLa cell as a template. Obtained DNA fragments were subcloned into pEGFP-C or -N vectors. The expression vectors for GFP- and FLAG-TRAF6[35] were a generous gift from Dr Jun-ichiro Inoue (University of Tokyo). The expression vectors for GST-GFP-nanobody[36] were a generous gift from Dr Yohei Katoh (Kyoto University). The expression vectors for GFP-rat LC3B[37] were a generous gift from Dr Tamotsu Yoshimori (Osaka University).

**Cell cultures and transfections.** A total of 293T (human embryonic kidney fibroblasts) cells and variety of MEFs (mouse embryonic fibroblasts) were cultured in Dulbecco's modified Eagle medium (DMEM, Nakarai) supplemented with 10% FCS (Gibco), 100 μg/ml gentamicin (Wako), and 60 μg/ml kanamycin (Wako). platE cell was maintained in DMEM/10% FCS with 1 μg/ml puromycin (Sigma) and 10 μg/ml blasticidin (Kaken pharmaceutical). Transfections were performed by using PEI MAX (Polysciences) or Lipofectamine LTX (Thermo Fisher Scientific) for 293T and platE, and Fugene 6 (Promega) for MEFs according to the manufacturers' protocols. MEFs-derived stable clones were cultured in DMEM/10% FCS with 1 μg/ml puromycin and/or 10 μg/ml blasticidin. FIP200 WT or KO was cultured previously[38]. ULK1/2 WT or DKO MEFs[39], Atg5 WT or KO MEFs[40], Nox1 WT or KO and Nox4 WT or KO MEFs[41], Atg16L1 WT or KO MEFs[42], and p62 WT or KO MEFs[43] were generous gift from Dr Craig B. Thompson (Memorial Sloan Kettering Cancer Center), Dr Noboru Mizushima (The University of Tokyo), Dr Denis Martinvalet (Genève University), and Drs Tatsuya Saitoh (Tokushima University) and Shizuo Akira (Osaka University), and Toru Yanagawa (University of Tsukuba).

**siRNA.** siRNAs were synthesized and duplexed by siRNA Co. Ltd. The siRNA sequences for mouse Rabs are shown in Supplementary Table 2. The siRNAs were transfected into cells by reverse transfection using Lipofectamine RNAi MAX (Thermo Fisher Scientific) according to the manufacturer's protocol. Knockdown efficiency was checked by RT-PCR using primer pair shown in Supplementary Table 2.

**Construction of *S. pneumoniae* mutant by allelic exchange mutagenesis.** Substitution of the *ply* gene in *S. pneumoniae* strain R6 was performed by double crossover recombination as described previously[14]. Briefly, Allele 1 *ply* gene of R6, or Allele 5 *ply* gene of ATCC6308, and subsequent *erm* cassette were joined with long 5′- and 3′-flanking regions that are homologous to the target locus were generated using two-step PCR as described previously[44] using the primers listed in Supplementary Table 3. PCR products were introduced into competent cells of the *S. pneumoniae* strain R6 as described previously[14], and transformants were selected by 1 μg/ml erythromycin. Substitution of the *ply* gene was confirmed by sequencing of PCR product amplified by primers listed in Supplementary Table 1.

**Recombinant retroviruses and infections.** The pMXs-puro and pMX-IRES-blast vectors were purchased from Cosmo Bio. Recombinant retroviruses were prepared as previously described[14]. In short, obtained retroviral plasmid was transfected into platE, and after 2 days culture supernatant containing retrovirus was collected and centrifugated, and clear supernatant was used for retroviral infection. For preparation of retrovirus for A549 cells, platE cells were cotransfected with retroviral plasmid and VSV-G plasmid. For retroviral infection, recipient cells were infected

with retroviruses by adding obtained supernatant of platE cells in the presence of polybrene for 6 h. Stable transformants were selected in DMEM/10% FCS with 1 μg/ml puromycin or 10 μg/ml blasticidin.

**Infection of cells with *S. pneumoniae* and sample preparation for fluorescence microscopy, western blotting, electron microscopy, and survivability assay.** Variety of MEFs/pIgR were infected with *S. pneumoniae* as previously described[14]. In short, MEFs seeded at $2 \times 10^5$ cells/well on glass coverslips in six-well plates were infected with fresh *S. pneumoniae* grown with THY broth at MOI (multiplicity of infection) of 100, and then centrifugated at 1000 rpm for 5 min at room temperature. Cells were incubated for 1 h at 37 °C in 5% $CO_2$ to allow bacterial uptake and invasion, washed with Hank's balanced salt solution (HBSS) three times, and DMEM/10% FCS with 200 μg/ml gentamycin and 200 U/ml catalase (Bacterial killing buffer) was added to each well and incubated for 30 min to kill extracellular bacteria. After changing the medium to DMEM/10% FCS with 100 μg/ml gentamycin and 200 U/ml catalase (incubation buffer), the cells were incubated for the indicated periods at 37 °C in 5% $CO_2$. If not stated otherwise, inhibitors were added in incubation buffer to avoid the influence on bacterial invasion efficiency. At each time point, the cells were fixed with 4% paraformaldehyde/PB (phosphate buffer) (Wako) for 15 min at room temperature. Fixed cells were washed with PBS three times and quenched with 50 mM $NH_4Cl$ in PBS for 10 min, permeabilized with 0.2% Triton X-100 in PBS for 10 min and blocked with 2% bovine serum albumin in TBS (Tris-buffered saline) for 30 min at room temperature. After staining using the indicated antibodies, the specimens were analyzed by confocal microscopy (Carl Zeiss LSM700). For western blot analysis, cells were lysed with 100 μl of 2× SDS sample buffer (Nakarai Tesque). Equal volumes of lysates were separated by SDS-PAGE and transferred to a PVDF membrane. For electron microscopy observation, cells seeded on 35 mm dishes were infected with *S. pneumoniae* as described above, fixed with 2% glutaraldehyde/4% paraformaldehyde in PBS for overnight at room temperature, post-fixed in 2% $OsO_4$, dehydrated with a graded ethanol series, and embedded in epoxy resin. Ultra-thin sections were stained with both uranyl acetate and lead citrate. For, intracellular bacterial survivability assay, MEFs seeded on 24-well plates were infected with wild-type of *S. pneumoniae* as described previously[14]. Briefly, MEFs were infected with *S. pneumoniae* strain R6 WT to allow bacterial invasion as described above, and were washed with HBSS twice, and were then incubated in bacterial killing buffer containing 200 μg/ml gentamicin for 15 min, and were incubated in bacterial killing buffer containing 200 μg/ml gentamicin and 10 μg/ml of penicillin G (Wako) for a further 15 min. After washing cells with HBSS three times, cells were incubated in incubation buffer for 1 h at 37 °C in 5% $CO_2$, and were then lysed using 1.0% saponin (Sigma-Aldrich), and lysates were serial diluted with 0.1% saponin in PBS and were plated onto THY-agar plates, and numbers of intracellular bacteria were counted and expressed in colony forming units (CFU).

**Quantification of cells with bacteria associated with autophagic markers.** We confirmed that infection efficiency was at least 100% when MEFs were infected with *S. pneumoniae* at MOI = 100. Thus, we evaluated PcLV formation activity by the percentage of cells containing PcV (*S. pneumoniae*-containing vesicles) associated with autophagic markers, such as GFP-LC3, p62-3Myc, ubiquitin etc., which was determined by visual counting with fluorescence microscopy. At least 500 PcVs, PcLVs, or *S. pneumoniae*-containing cells were examined in triplicate for each experimental condition. Error bars indicate the standard errors of the means.

**Purification of GST-GFP-nanobody protein.** E. coli BL21 harboring pGST-GFP-nanobody was cultured in L-broth with 50 μg/ml of ampicillin for 2 h at 37 °C, then isopropyl-1-thio-β-D- galactopyranoside was added to the culture medium at a final concentration of 0.1 mM. After incubation for 2 h at 37 °C, bacteria were harvested, and GST fusion proteins were purified according to the manufacturer's protocol using glutathione Sepharose 4B (GE Healthcare).

**GST pull-down assay using 293T cell lysates.** A total of 293T cells transfected with indicated plasmids seeded onto six-well plate were suspended in 500 μl of wash buffer (TBS/5 mM $MgCl_2$, 1 mM EDTA, 0.05% NP-40) including complete protease inhibitor cocktail (Roche), chilled on ice for 30 min, and centrifuged at $16,900 \times g$ at 4 °C for 10 min. Five microliters of GST-GFP-nanobody bound to glutathione Sepharose 4B was mixed with cleared lysates of the 293T cells and incubated with rotation for 2 h at 4 °C. After incubation, the beads were washed with 1 ml of wash buffer three times, and the bound proteins were analyzed by immunoblotting using indicated antibodies.

**Immunoprecipitation using 293T cell lysates.** A total of 293T cells transfected with indicated plasmids seeded onto six-well plate were suspended in 500 μl of wash buffer (TBS/5 mM $MgCl_2$, 1 mM EDTA, 0.05% NP-40) including complete protease inhibitor cocktail (Roche), chilled on ice for 30 min, and centrifuged at $16,900 \times g$ at 4 °C for 10 min. Five microliters of GFP-Trap beads (Chromotek) was mixed with cleared lysates of the 293T cells and incubated with rotation for 2 h at 4 °C. After incubation, the beads were washed with 1 m of wash buffer three times, and the bound proteins were analyzed by immunoblotting using indicated antibodies.

**Statistics and reproducibility**. At least 500 PcVs, PcLVs, or pneumococcus-containing cells were examined in triplicate, in each experiment. Data are the means ± SEM. *P* values were calculated using Student's *t* test. If SD is significantly different ($F < 0.05$), *P* values were calculated using Mann–Whitney *U* test by using Prism6.

**Reporting summary**. Further information on research design is available in the Nature Research Reporting Summary linked to this article.

## Data availability

The authors declare that all data supporting the findings of this study are available within the article and its supplementary information files. All source data underlying the graphs presented in the main figures are made available in Supplementary Data 1.

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

## Acknowledgements

We thank Drs Masaaki Komatsu, Craig B. Thompson, Noboru Mizushima, Denis Martinvalet, Tatsuya Saitoh, Shizuo Akira, and Toru Yanagawa for providing reagents. We thank Mr Tomohiro Nakai for technical supports. This work was supported by Grant-in-Aid for Scientific Research (C) (19K07568, 16K08800, 25460555) to Mi.O. from the Ministry of Education, Culture, Sports, Science and Technology (MEXT). This work was supported by grants from the Naito Foundation and the Uehara Foundation.

## Author contributions

Mi.O. supervised the overall research project. Mi.O., N.T., S.S., and M.T. performed the experiments. B.C. provided bacterial strains. M.Y., S.K., and I.T. performed EM experiment. Mi.O., N.T., S.S., M.T., A.R., J.G., H.T. and Ma.O. analyzed the data. Mi.O. wrote the manuscript with input from all co-authors.

## Competing interests

The authors declare no competing interests.
