## [Peer review file · Communications Biology]

Reviewer #2 (Remarks to the Author):

The manuscript by Ogawa et al investigates the interaction with *S. pneumoniae* (Sp) with the LC3-associated phagocytosis (FIP200 independent) and canonical (FIP200-dependent) autophagy pathways. The authors provide data to support their conclusion that pneumolysin promotes uptake of Spn into LC3-pos vacuoles which are permissive for bacterial survival, and that these processes are dependent on the cytolytic activity of pneumolysin. The authors then provide evidence that Sp uptake via LAP precedes destruction via canonical autophagy, and suggest that this is a coordinated cellular process that involves reprogramming the LAP vacuoles to the classical autophagic vacuoles. The authors propose a novel pathway of maturation of the PclAP to PcAV via the loss of NDP52 and LC3 and the subsequent recruitment of LC3. I don't think this case has been made sufficiently, however. The results presented could be explained by a simpler model whereby Sp invades via LAP, followed by escape into the cytosol where they are recruited to canonical autophagy via Ub, P62 and LC3. Timecourse TEM experiments with quantitation, coupled with immunofluorescence to define the association of different autophagic markers (p62, NDP52, Gal8, Ub etc) could resolve this possibility. Finally, some of the assays chosen are not ideal, particularly with the quantitation of cells with each structure (PclAPs) rather than the number of each structure when a very high multiplicity of infection was chosen for these experiments (see comment below).

Major Concerns

1. The role of LAP in Spn uptake is very interesting, but curious why no quantitative measures of Spn uptake were undertaken (i.e. Gentamicin-protection assays, or microscopic quantitation following differentiation of extracellular vs intracellular bacteria).
2. Given that this manuscript is attempting to define a temporal process of invasion, and sequential formation of PclAP, LdPclAP and PcAV structures, detailed timecourse experiments with careful quantitation are essential.
3. The TEM figures in particular have no quantitation associated with them. This is essential if the reader is to believe that the presented figures are representative.
4. This is a very difficult paper for a reader not intimately informed with the mechanistic of autophagy cell biology and biochemistry. For example, in Fig 2K, there is no explanation for why "si Atg14L" is used. Likewise, the experiments in Figs 3C, E and G were quite difficult to follow. What was the reason for investigating Myo6 in Fig 3G? While I appreciate that these are complex cellular processes that are under investigation, I think some expanded explanation and/or diagrammatic clarification of processes would help a less-informed reader to follow the flow of ideas and experiments.
5. There are no details/reference explaining the Sp survival assays. How are these performed?
6. In Fig 3A, could these results also be explained by NDP52 labeling the damaged vacuole (via Gal8), and p62 labeling the bacterial surface? What is the evidence that the vacuole is modified rather than Sp moving from a damaged vacuole to the cytoplasm? Perhaps some quantitative TEM of the LdPclAP-positive cells would resolve this? Would these p62-pos bacteria be membrane-enclosed or cytosolic?
7. The method of quantitation of LC3-positive bacteria is not ideal in my opinion. With a very high MoI for these infections (MoI=100), it is quite likely that more than one bacteria will invade each cell. However, by quantitating the number of LC3-positive bacteria containing cells (ie. cell count) rather than LC3-pos bacteria (bacteria count), important differences might be missed. One example of this is in Fig 3D 1h panels – there are 9 visible LC3-positive bacteria in the si Luc panels, and only 3 LC3-positive bacteria in the si Atg14L panels, a 3-fold difference, yet the quantitation of these in Fig 3C 1h panel suggests there is no difference.
8. There is no quantitation for the colocalization of Atg16L, Lc3 and p62 in Fig 5B.
9. I am confused by the results in Fig 5A – it seems like the level of IP for the Δ WD (lane 5) is very similar to the FL (lane 1). How then is the WD domain responsible for P62 binding as stated on lines 289-291?

Questions:

1. Is LC3 recruited to the PclAPs during internalisation, or is it recruited to intracellular vacuoles (i.e. LC3 earlier in Fig 3H than is indicated) – similar to that described for *Trypanosoma cruzii*
2. General conclusions about the role of these processes in Sp disease would be stronger if critical

findings were replicated in humal nasopharyngeal cells. All of these results were obtained with MEFs, a cell type (fibroblast) and host (mouse) that is not naturally infected during Sp disease. I think it would be really nice if the critical findings could be replicated in a cells derived from the human nasopharynx.

3. Related to 2 – there is a lot of extrapolation of findings between different model systems, without performing the appropriate controls that would validate these statements. For example, the role of the WD region on Shigella clearance (lines 199-200).

4. In Fig 1B, would the 2h WT timepoint show a higher number of PclAPs due to reduced degradation through PcaVs?

5. What is the evidence that PcaVs lead to Sp killing?

6. In Fig 3H, if FIP200 and Beclin1 are both required for PclAP → LdPclAP transition, why is there an effect of Beclin1 siRNA in Fig 3F?

Minor comments;

1. Statement that PclAPs don't have bactericidal activity should be determined by a timecourse survival assay (e.g. Gm protection).

2. Where is the actual data for Fig S2A?

3. Lines 176-177 – this should be verified by a control assay. How do you know that your cells are behaving normally?

4. Fig 3F – 2h black bar should be "si Luc"?

5. Line 259 – Ogawa reference not formatted properly by Endnote

6. Line 263 – "E1 inhibitor" should be defined. What is E1?

7. Fig 4 – the panels that are using FIP200 KO MEFs should be annotated as such (i.e. as done for Figs 1-3). As presented it gives the impression that WT cells are being used.

8. In Fig 4G, it seems much more likely that Gal8 is linking NDP52 to the PclAP, rather than the other way around. Is p62 and Ub bound to the PclAP, or is p62 bound directly to Sp via Ub?

Reviewer #3 (Remarks to the Author):

Comments for Author

The manuscript by Dr Ogawa and colleagues describes the processes involved in hierarchical autophagy induction by intracellular pneumococci. The study is novel, well designed and identifies the autophagy proteins triggered by intracellular pneumococci. The experiments support the conclusions of the study, but I have few concerns that need to be addressed.

Major comment:

1. The experiments are done in mouse embryonic fibroblast cell lines, which are not the ideal host for pneumococci in vivo. It would be interesting if the authors could verify the critical results in cells that naturally encounter pneumococci during an infection such as lung epithelial cells and/or immune cells like macrophages. Do pneumococci also form PclAPs and PcaVs in these cell types?

Minor comments:

1. In Fig. 1B, the data for LC3B associated bacteria at 2h time point in WT control is missing.

2. In Figs. 1 and 2, the data for LC3 and lysotracker associated pneumococci in control wild type MEFs cells is missing.

3. Fig. 2H, Bacterial killing is misspelt.

4. In Fig. 2, the panels could be ordered for the reader to sequentially follow the results. Right now, panel 2E is located below A.

Dear Reviewers:

We sincerely appreciate the helpful comments and suggestions provided by you with regard to our manuscript. We have read carefully all of your comments and have responded accordingly. Our detailed responses are mentioned below, including additional experimental data where appropriate. Revisions are highlighted in blue in the revised manuscript.

We hope that you find these revisions satisfactory.

Reviewer #1 (Remarks to the Author):

This was an interesting study that provided information on what happens intracellularly for pneumococcal infections. The data is well presented; however, some modifications could help with overall clarity of the manuscript.

Thank you for the positive comment.

1. For figure 1C, please add arrows

As suggested, we have added arrows in Fig. 1C.

2. Line 97: Please indicate the pneumococcal strain here

As suggested, we have added strain name R6 in the revised manuscript.

3. Line 103: Please indicate the pneumococcal strain here (to make the statement in 144 seem clear)

As suggested, we have added strain name R6 in the revised manuscript.

4. Line 108: Define leaky (is there some measure of this? A rate?) I am not suggesting doing that experiment, but please support this statement as leaky is vague.

We apologize for the confusion. Although we observed that PcV in Atg5 KO MEFs is less spacious than PcLV in FIP200 KO MEFs, we could not estimate it. We have replaced the TEM image of Atg5 KO MEFs by FIP200 WT MEFs for comparison of

PcLV with PcAV, and we have corrected sentences in the text (please see lines 112–113 and Fig. 1C).

5. A note on abbreviations, I think LAP is well established but further adding Pc for pneumococcus containing and having the P be capital is a bit odd throughout. The V is included in the PcAV abbreviation and it is curious why it isn't included in the LAP one. If space is not an issue, why not just say "pneumococcus containing LAP vacuoles." Further saying LAPs for LAP vacuoles and not having the V there causes a lack of clarity throughout. Also the fact that V can be for vesicles and vacuoles. In the figure it is fine, but in text it is difficult to follow. It is further complicated by the pc in the vector names

We thank the reviewer for this helpful comment. As suggested by the reviewer, we have changed PcLAP into PcLV and renamed LC3-associated phagosomes as LAPosomes because LC3-associated phagocytosis (LAP) is a process but not a vesicle. "P" is commonly used in nomenclature for pneumococcus-related proteins and events (i.e., PspA, PcpA, PsaA, PhtA, and IPD), and c, meaning "containing," is also commonly used for xenophagy, such as GAS-containing autophagic vacuoles and Salmonella-containing vacuoles. Furthermore, we have already designated pneumococci-containing autophagic vacuoles as PcAV in our previous paper. Thus, we decided to use PcLV in this paper.

6. Is there direct evidence that the ply mutant has been complemented successfully? Please state here

We have added arrows to indicate successful Ply complementation in Fig. 1K, which was confirmed by immunostaining with a specific antibody against pneumolysin. Furthermore, we have corrected the text to ensure clarity (please see lines 146–147).

7. Line 162: please include full name of WD domain before proceeding

We have clarified the full name of WD as "structural domain composed of approximately 40 amino acids, often terminated by a tryptophan–aspartic acid" (please see lines 167–169).

8. Please put the (appreciated) model at end of figure 2 ordering as was done in figure 3.

Thank you for this comment. We have placed the new schematic diagram at the end of Fig. 2 as Fig. 3 (please see Fig. 2N).

9. Line 231: This does not need an abbreviation, please spell this out.

Thank you for this helpful comment. As suggested, we have replaced *LdPcLAP* with “*NDP52-delocalized PcLV*.”

10. The blots in figure 5 are hard to follow.

We have clarified and separated Fig. 5A into Fig. 6A and Fig. 6C and revised the schematic diagram in Fig. 6J and Fig. S9.

11. Please change the color of the poly ubiquitin chain as it looks like the bacteria of previous models.

Thank you for this comment. As suggested, we have changed the color of the polyubiquitin chain.

12. Line 758: Diagram of f?

Thank you for this comment. We have corrected this accordingly.

Reviewer #2 (Remarks to the Author):

The manuscript by Ogawa et al investigates the interaction with *S. pneumoniae* (Sp) with the LC3-associated phagocytosis (FIP200 independent) and canonical (FIP200-dependent) autophagy pathways. The authors provide data to support their conclusion that pneumolysin promotes uptake of Spn into LC3-pos vacuoles which are permissive for bacterial survival, and that these processes are dependent on the cytolytic activity of pneumolysin. The authors then provide evidence that Sp uptake via LAP

precedes destruction via canonical autophagy, and suggest that this is a coordinated cellular process that involves reprogramming the LAP vacuoles to the classical autophagic vacuoles. The authors propose a novel pathway of maturation of the PcLAP to PcAV via the loss of NDP52 and LC3 and the subsequent recruitment of LC3. I don't think this case has been made sufficiently, however. The results presented could be explained by a simpler model whereby Sp invades via LAP, followed by escape into the cytosol where they are recruited to canonical autophagy via Ub, P62 and LC3. Timecourse TEM experiments with quantitation, coupled with immunofluorescence to define the association of different autophagic markers (p62, NDP52, Gal8, Ub etc) could resolve this possibility. Finally, some of the assays chosen are not ideal, particularly with the quantitation of cells with each structure (PcLAPs) rather than the number of each structure when a very high multiplicity of infection was chosen for these experiments (see comment below).

Thank you for these insightful comments to strengthen our manuscript. We have performed additional experiments to clarify points raised by the reviewer as described below.

Major Concerns

1. The role of LAP in Spn uptake is very interesting, but curious why no quantitative measures of Spn uptake were undertaken (i.e. Gentamicin-protection assays, or microscopic quantitation following differentiation of extracellular vs intracellular bacteria).

As suggested by the reviewer, we have examined the relationship of invasion efficiency with LAP activity. As shown below, invasion efficiency was not affected by the LAP activity (left panel below). We are speculating that the dynamics of LAP in phagocytes and PcLV in non-phagocytic cells are not the same. We have added these results in the revised MS (Fig. S1Q) (please see lines 156–157). Next, we conducted a time course experiment of PcLV formation and found that LC3 deposition did not occur until after 60 min of infection (right panel below). This finding clearly suggests that the invasion event is distinct from PcLV formation. Together, we assume that PcLV activity has no effect on pneumococcal invasion efficiency, and vice versa. We have added these results in the revised MS (Fig. 4A).

2. Given that this manuscript is attempting to define a temporal process of invasion, and sequential formation of PcLAP, LdPcLAP and PcAV structures, detailed time course experiments with careful quantitation are essential.

Thank you for this helpful comment to strengthen our paper.

We performed detailed time course experiments to define PcLV, NDP52-delocalized PcLV, and PcAV. Please see below. We have added these data in the revised MS (please see lines 275–306 and Figs. 4A–C).

*We presumed hierarchical autophagy processes through NDP52-delocalized PcLV transition in *S. pneumoniae*-infected cells occurred like this (please see below). We have added these schematic models in the revised MS for clarity (Fig. S6).*

3. The TEM figures in particular have no quantitation associated with them. This is essential if the reader is to believe that the presented figures are representative.

Although we are certain that PcV in Atg5 KO MEFs is less spacious than PcLV in FIP200 KO MEFs, we could not estimate it. We have replaced the TEM image of Atg5 KO MEFs by FIP200 WT MEFs for comparison of PcLV with PcAV. We have corrected the sentences in the text (please see lines 111–113 and Fig. 1C).

4-1. This is a very difficult paper for a reader not intimately informed with the mechanistics of autophagy cell biology and biochemistry.

For example, in Fig 2K, there is no explanation for why “si Atg14L” is used.

We used siAtg14L to exclude Atg14L-dependent canonical autophagy. We have added explanation in the text and figures (please see line 208 and Figs. 2H, 2J, and S8E-F).

4-2. Likewise, the experiments in Figs 3C, E and G were quite difficult to follow.

We have added extensive explanation in the text and additional figures, and we have revised the schematic diagram (please see lines 243–262 and Figs. 3E–G and J).

4-3. What was the reason for investigating Myo6 in Fig 3G?

The Myo6–NDP52 interaction is involved in tethering NDP52-positive vacuoles with lysosomes. As described above, we have added sentences in the text (please see lines 267–271).

4-4. While I appreciate that these are complex cellular processes that are under investigation, I think some expanded explanation and/or diagrammatic clarification of processes would help a less-informed reader to follow the flow of ideas and experiments.

Thank you for this comment. As described above in 4-1, 4-2, and 4-3, we have added this explanation in the manuscript.

5. There are no details/reference explaining the Sp survival assays. How are these performed?

As we cited in our previous report in the original MS, we have added the brief protocol in the Methods part of our revised MS (please see lines 616–624).

6-1. In Fig 3A, could these results also be explained by NDP52 labeling the damaged vacuole (via Gal8), and p62 labeling the bacterial surface?

To investigate whether the NDP52–Gal8 interaction is the critical event for PcLV formation, we examined PcLV formation and NDP52 deposition on pneumococci-containing vacuoles in the presence or absence of the ubiquitin-activating enzyme E1 inhibitor (PYR-41). As shown below, NDP52 recruitment was robustly suppressed by PYR-41 treatment, which supports our notion that poly-Ub, p62, NDP52, and Atg16L WD are mutually dependent for their recruitment to intracellular pneumococci and that these proteins are required for PcLV formation. We have added the data in the revised MS (please see lines 342–344 and Fig. 5B).

To investigate whether p62 is deposited on the bacterial surface or the membranes of bacteria-containing vacuoles, we investigated the localization of LC3, p62, poly-Ub, and bacteria. As shown below, the entire signal was clearly deposited on membranous structures engulfing pneumococci. We have added the data in the revised MS (please see lines 359–363 and Fig. 5H).

*Notably, as shown below, we rarely observed p62 and poly-Ub deposited on bacteria. We speculated that when a small population of *S. pneumoniae* escapes from endosomes into the cytosol, the bacteria would be stained like this. This staining pattern is reminiscent of GAS-targeting xenophagy reported by Dr. Nakagawa's group. We are speculating that quite a small population of pneumococci can evade from endosomes into the cytosol and most bacteria still stay in damaged membrane compartments.*

6-2. What is the evidence that the vacuole is modified rather than Sp moving from a damaged vacuole to the cytoplasm?

Thank you for this insightful comment. We know that it has been recently reported that S. pneumoniae can escape from autophagosomes into the cytosol in human brain endothelial cells (Ref. 31). As suggested by the reviewer, we examined whether intracellular pneumococci can escape into the cytosol at the NDP52-delocalized PcLV stage after 2 h of infection in our experimental setting. As shown below, p62 and poly-Ub were not deposited on individual bacterium, but they localized on ring-shaped vesicle membranes at the PcLV and NDP52-delocalized PcLV stage, suggesting that most bacteria are engulfed in membranous structures. Notably, when the intracellular membrane was stained with GFP-Lact-C2, a marker for phosphoserine on endosomes, at the NDP52-delocalized PcLV stage, most of the bacteria existed in the GFP-Lact-C2-labeled vacuoles. We have added these results in the revised MS. Furthermore, we previously reported that PcAV observed at 2 h p.i. is frequently decorated with Galectin-3, strongly suggesting that most pneumococci were enclosed by endosome-derived damaged membranes even at the subsequent PcAV stage. Together, we presumed that most pneumococci remained engulfed in the membrane compartment in our setting (please see lines 359–367 and Figs. 5H–I).

6-3. Perhaps some quantitative TEM of the LdPcLAP-positive cells would resolve this?

Thank you for this helpful advice. We cannot use correlative light and electron microscopy, but we are planning to investigate the sequential observation of PcLV to PcAV by immune EM.

6-4. Would these p62-pos bacteria be membrane-enclosed or cytosolic?

As described above, in our experimental setting, we presumed that most intracellular S. pneumoniae appeared to be membrane enclosed (please see Fig. 5H).

7-1. The method of quantitation of LC3-positive bacteria is not ideal in my opinion. With a very high MoI for these infections (MoI=100), it is quite likely that more than one bacteria will invade each cell.

In the case of Salmonella, invasion efficiency was quite high, and we infected cells with Salmonella at moi = 5, but for S. pneumoniae R6 and Shigella case, invasion efficiency was quite low (approximately 3% and 0.01%). Thus, invasion efficiency is strictly dependent on the bacterial species and strains. In our experimental setting, we infected cells with S. pneumoniae at an MOI of 100. However, MOI in actual infections (invaded bacteria/cell) is estimated to be 3 at maximum.

7-2. However, by quantitating the number of LC3-positive bacteria containing cells (ie. cell count) rather than LC3-pos bacteria (bacteria count), important differences might be missed. One example of this is in Fig 3D 1h panels – there are 9 visible LC3-positive bacteria in the si Luc panels, and only 3 LC3-positive bacteria in the si Atg14L panels, a 3-fold difference, yet the quantitation of these in Fig 3C1h panel suggests there is no difference.

Thank you for this comment. We agree and understand that it is better to count individual LC3-positive bacterium and estimate the frequency of LC3-positive bacteria relative to that of the total invaded bacteria. However, similar to other Streptococci, S. pneumoniae is a coccus and present as diplococci and chained cocci, and, furthermore, intracellular pneumococci are observed in various morphologies. For Shigella and Salmonella infections, because each bacterium is surrounded by an individual vacuole, we can easily count the number of bacteria inside vacuoles. Streptococcus bacterium share the cell wall with adjacent bacterial cells. Therefore, the number of pneumococci by counting bacterial cells did not properly reflect the absolute number of bacteria. We tried to quantify the frequency of marker-negative pneumococci to estimate the total number of pneumococci, but we could not quantify them. Therefore, we decided that, for the estimation of intracellular dynamics of pneumococci, counting cells with marker-positive pneumococci was more reliable and reasonable. Therefore, we showed the percentage of cells with marker-positive pneumococci.

8. There is no quantitation for the colocalization of Atg16L, LC3 and p62 in Fig 5B.

In S. pneumoniae infection, Atg5–Atg16L1 recruitment was quite transient and quantitation of Atg16L, LC3, and p62-colocalization was quite difficult. Furthermore, as we observed samples by four colors in Fig. 5B (Fig. 6B in the revised MS), Cy3 (for endogenous p62) staining made it more difficult to quantify their colocalization.

9. I am confused by the results in Fig 5A – it seems like the level of IP for the Δ WD (lane 5) is very similar to the FL (lane 1). How then is the WD domain responsible for P62 binding as stated on lines 289-291?

Thank you for this comment. We have replaced Fig. 5A and separated it into two figures (please see Figs. 6A and C).

Questions:

1. Is LC3 recruited to the PcLAPs during internalisation, or is it recruited to intracellular vacuoles (i.e. LC3 earlier in Fig 3H than is indicated) – similar to that described for *Trypanosoma cruzii*

*Thank you for this comment. As described above, LC3 was not recruited at 30 min p.i., and PcLV was acutely induced at 60 min p.i. Thus, we are assuming that PcLV was different from LAP by *Trypanosoma cruzi* in our experimental setting.*

2. General conclusions about the role of these processes in Sp disease would be stronger if critical findings were replicated in human nasopharyngeal cells. All of these results were obtained with MEFs, a cell type (fibroblast) and host (mouse) that is not naturally

infected during Sp disease. I think it would be really nice if the critical findings could be replicated in a cells derived from the human nasopharynx.

Thank you for this helpful comment to strengthen our MS. We have added data showing that NDP52-localized PcLV transiently appeared and disappeared in FIP200 knockdown A549 cells (human lung epithelial cell line), and antioxidants, such as NAC and apocynin, had no effect on PcLV formation. Furthermore, we constructed A549 cells stably expressing mouse Atg16L1 (KD resistant) and performed a human Atg16L1 knockdown experiment. We found that Atg16L1 WD-rescued A549 cells were deficient in PcLV and PcAV (NDP52 negative but LC3 positive). These results show that the findings observed in the MEF cells can be extended in human lung epithelial cells. We have added these results in our revised MS (please see lines 307–334 and Figs. 4D–G and S7).

3. Related to 2 – there is a lot of extrapolation of findings between different model systems, without performing the appropriate controls that would validate these statements. For example, the role of the WD region on Shigella clearance (lines 199-200).

Thank you for this comment. We agree and understand that it is better to set control experiments using the same pathogens used in previously paper. However, each bacteria and strains are recognized by distinct xenophagic mechanisms, and they have a distinct strategy to evade from them. Under these conditions, putting the same bacterial strains as a control is very difficult. In the case of Salmonella-induced LAP, we used almost the same experimental setting as Dr. Yoshimori's group by using

Atg16L1 KO MEFs constructed by Dr. Akira. As for Shigella, Dr. Shao's group recently showed similar results with those of Dr. Xavier's group (Refs. 24 and 25) showing WD domain's relevance in Shigella targeting noncanonical xenophagy.

4. In Fig 1B, would the 2h WT timepoint show a higher number of PcLAPs due to reduced degradation through PcAVs?

Thank you for this comment. We have added the data of WT at 2 h p.i. in Fig. 1B. At this time point, PcAV already occurred in WT. Thus, the population of LC3-positive PcV (pneumococci-containing vacuoles) containing cells remains high. However, the abundance of NDP52-positive vacuoles dramatically reduced at 2 h p.i. in WT MEFs. Notably, as NDP52-delocalized PcLV transition proceeded in WT MEFs, even in Atg14L-knockdown conditions, we posited that PcAV activity is not involved in NDP52-delocalized PcLV transition.

5. What is the evidence that PcAVs lead to Sp killing?

Thank you for this comment. We have previously reported that PcAV in WT MEFs exerts bactericidal effect on intracellular S. pneumoniae (Ogawa M. et al., Molecular mechanisms of Streptococcus pneumoniae-targeted autophagy via pneumolysin, Golgi-resident Rab41, and Nedd4-1-mediated K63-linked ubiquitination. Cell Microbiol 20, e12846 (2018)).

6. In Fig 3H, if FIP200 and Beclin1 are both required for PcLAP → LdPcLAP transition, why is there an effect of Beclin1 siRNA in Fig 3F?

We presumed that Atg14L KD and Beclin1 have the same effect on PcLV → NDP52-delocalized PcLV transition as they both are components of the PI3KC3 complex. Thus, if each is absent in FIP200 KO MEFs, then PcLV → NDP52-delocalized PcLV transition would be impaired. Thus, we clarified in the MS that either FIP200 or PI3KC3 component is required. As suggested, we have revised Fig. 3H (Fig. 3J in the revised MS) and Fig. 4H.

Minor comments;

1. Statement that PcLAPs don't have bactericidal activity should be determined by a time course survival assay (e.g. Gm protection).

Thank you for this comment. The results of the time course experiment reveal that PcLVs show acute peak at 1 h p.i., and it sequentially proceeded into NDP52-delocalized PcLV at 2 h p.i. Thus, we believe that our experimental design is appropriate.

2. Where is the actual data for Fig S2A?

Thank you for this comment. It is a literature review from Dr. Yoshimori's and Dr. Florey's papers. We used almost the same experimental setting as Dr. Yoshimori's group by using Atg16L1 KO MEFs constructed by Dr. Akira. We have added two references in Fig. S2A.

3. Lines 176-177 – this should be verified by a control assay. How do you know that your cells are behaving normally?

Thank you for this comment. We employed almost the same experimental setting as Dr. Yoshimori's group by using Atg16L1 KO MEFs constructed by Dr. Akira. Furthermore, we confirmed that canonical autophagy normally proceeds by chloroquine assay (please see Figs. S2C–D) and verified the complementation of FL and Δ WD through Western blotting. In addition, we performed experiments using human lung epithelial cells, which showed similar results as in Atg16L1 KO MEFs complemented with FL and Δ WD of Atg16L1.

4. Fig 3F – 2h black bar should be “si Luc”?

Thank you for this comment. We have fixed it.

5. Line 259 – Ogawa reference not formatted properly by Endnote

Thank you for this comment. We have fixed it.

6. Line 263 – “E1 inhibitor” should be defined. What is E1?

Thank you for this comment. We have added experiments about the E1 inhibitor as an ubiquitin-activating enzyme inhibitor (please see line 342).

7. Fig 4 – the panels that are using FIP200 KO MEFs should be annotated as such (i.e. as done for Figs 1-3). As presented it gives the impression that WT cells are being used.

Thank you for this helpful comment. We have fixed it.

8. In Fig 4G, it seems much more likely that Gal8 is linking NDP52 to the PcLAP, rather than the other way around. Is p62 and Ub bound to the PcLAP, or is p62 bound directly to Sp via Ub?

As described above, we are assuming that in our experimental setting, poly-Ub, p62, NDP52, and Atg16L WD are mutually dependent for their recruitment to intracellular pneumococci and that Gal8–NDP52 interaction is important but not utmost in this complicated process. Furthermore, as shown in Fig. 5G, although NDP52 recruitment dramatically decreased to the 4% level of siLuc via Gal8 knockdown, LC3 deposition remained at the 57% level of siLuc. This result also supported our notion.

Furthermore, as described above, we investigated whether p62 is localized to the bacterial surface or bacteria-containing vacuoles. When the localization of LC3, p62, poly-Ub, and bacteria were examined, all signals were clearly deposited on the membrane of bacteria-containing vacuoles.

As described above, if bacteria are directly decorated by poly-Ub and p62, then it will look like below, which is rarely observed. However, in our experimental setting, most PcLVs, NDP52-delocalized PcLVs, and PcAVs looked as ring-shaped structures, and individual bacterium is not deposited by poly-Ub and p62.

Reviewer #3 (Remarks to the Author):

Comments for Author

The manuscript by Dr Ogawa and colleagues describes the processes involved in hierarchical autophagy induction by intracellular pneumococci. The study is novel, well designed and identifies the autophagy proteins triggered by intracellular pneumococci. The experiments support the conclusions of the study, but I have few concerns that need to be addressed.

Thank you for the positive comment.

Major comment:

1. The experiments are done in mouse embryonic fibroblast cell lines, which are not the ideal host for pneumococci in vivo. It would be interesting if the authors could verify the critical results in cells that naturally encounter pneumococci during an infection such as lung epithelial cells and/or immune cells like macrophages. Do pneumococci also form PcLAPs and PcAVs in these cell types?

Thank you for the helpful comment to improve our MS. We have added data showing that NDP52-localized PcLV transiently appeared and disappeared in FIP200 knockdown A549 cells (human lung epithelial cell line), and antioxidants, such as NAC and apocynin, had no effect on PcLV formation in A549 cells. Furthermore, we constructed A549 cells stably expressing mouse Atg16L1 (KD resistant) and performed human Atg16L1 knockdown experiments. We found that Atg16L1 WD-rescued A549 cells were deficient in PcLV and PcAV (NDP52 negative but LC3 positive). This population appeared in FL-rescued cells at 2 h p.i. These results clearly show that our findings observed in MEF cells can be replicated in human lung epithelial cells. We have added these results in the revised MS (please see lines 307–334 and Figs. 4D–G and S7).

Pictures of PcAVs in A549 cells is shown below.

Minor comments:

1. In Fig. 1B, the data for LC3B associated bacteria at 2h time point in WT control is missing.

Thank you for this comment. We have added the data of WT at 2 h p.i. in Fig. 1B. At this time point, PcAV already occurred in WT. Thus, the population of LC3-positive PcV (pneumococci-containing vacuoles) containing cells remains high. However, the abundance of NDP52-positive vacuoles dramatically reduced at 2 h p.i. in WT MEFs.

2. In Figs. 1 and 2, the data for LC3 and lysotracker associated pneumococci in control wild type MEFs cells is missing.

Thank you for the comment. We previously reported the results in WT MEFs. As shown below, also in WT MEFs, PclVs were not acidified in S. pneumoniae WT-infected cells, although vacuoles containing Δ ply bacteria were frequently acidified. We have added sentences to the text (please see lines 149–150).

3. Fig. 2H, Bacterial killing is misspelt.

Thank you for this comment. We have fixed it.

4. In Fig. 2, the panels could be ordered for the reader to sequentially follow the results. Right now, panel 2E is located below A.

Thank you for this comment. We have fixed it.

Reviewers' comments:

Reviewer #1 (Remarks to the Author):

All of my comments/concerns have been adequately addressed.

Reviewer #2 (Remarks to the Author):

The revised manuscript by Ogawa et al investigates the interaction with *S. pneumoniae* (Sp) with the LC3-associated phagocytosis (FIP200 independent) and canonical (FIP200-dependent) autophagy pathways. In response to my previous review of this manuscript, the authors have performed a number of additional experiments to address my previous concerns, and are to be commended for undertaking these additional experiments. The manuscript is greatly improved as a result. I have a few additional concerns/queries below. My major concern is the use of penicillin in the Sp invasion experiments, as this antibiotic choice would also kill bacteria localised to the cytoplasm and damaged vacuoles. This raises the concern that the process being described is specific to antibiotic-killed bacteria.

1. (previous comment 2) In Fig 4, it seems a bit confusing to use two different colour schemes in Figs 4B and C. Suggest standardising the colour of the LC3 bars.

2. (previous comment 2) In Fig 4B, I am confused as to why LC3, p62 and NDP52 was not included for both of panels 4B and 4C. If the model (in 4F) is correct, then NDP52 would stay associated with p62 staining in the FIP200-KO MEFs? For clarity, I think that P62 staining should be included in panel 4B, and NDP52 staining should be included in panel 4C.

3. (previous comment 4-3) Line 2 – should be "Upon knockdown of myosin VI"?

4. (previous comment 5) The use of penicillin G for bacterial invasion experiments to kill extracellular bacteria (lines 615-623) is a major concern, as penicillin efficiently enters the cell and kills bacteria in the cytoplasm and probably damaged endosomal compartments (see PMIDs 2501788 and 24331465, for example). This may account for the very low invasion rate (1-2%) seen in Fig S1Q. The major concern with this use of penicillin is that the autophagic process being described in this manuscript may be targeting penicillin-killed pneumococcus, rather than viable pneumococcus. To rule out this possibility, this experiment should be repeated, but using gentamicin to kill extracellular bacteria.

5. (previous comment 7-1). I disagree with the notion that very few bacteria are invading into each cell, especially in light of my concern with using penicillin to kill extracellular bacteria (above). In multiple figures it can be observed that multiple bacteria have invaded individual cells (e.g. Fig 1A, at least 4 Sp invasions; Fig 1K, at least 5; Fig 3D, at least 7 for siLuc and 6 for Atg14L). My previous comment stands as a point of concern that should be addressed.

6. (previous comment 7-2). I agree that counting chains could be challenging. Given that a chain of cocci would have arisen from a single invasion event, could each bacterial chain be counted as a single bacterium. i.e. quantitate as NDP52-pos chains etc.

Reviewer #3 (Remarks to the Author):

The authors have addressed most of my comments. However, I still have a minor comment/clarification:

1. In Fig. S7A,B the authors mention that "Notably, at 3 h p.i., LC3 recruitment remained high in siLuc-treated cells.

311 However, it was significantly decreased in siFIP200-treated cells, suggesting that subsequent PcAV is induced in siLuc-treated cells at 3 h p.i". But the data comparing the siLuc-treated cells vs

siFIP200-treated cells are on two different plots. Moreover, there is no statistical significance reported in the figures.

Dear Reviewers:

We sincerely appreciate the helpful comments and suggestions provided by you with regard to our manuscript. We have read carefully all of your comments and have responded accordingly. Our detailed responses are mentioned below, including additional experimental data where appropriate. Revisions are highlighted in **blue** in the revised manuscript.

We hope that you find these revisions satisfactory.

Reviewers' comments:

Reviewer #1 (Remarks to the Author):

All of my comments/concerns have been adequately addressed.

Thank you for the positive comment.

Reviewer #2 (Remarks to the Author):

The revised manuscript by Ogawa et al investigates the interaction with *S. pneumoniae* (Sp) with the LC3-associated phagocytosis (FIP200 independent) and canonical (FIP200-dependent) autophagy pathways. In response to my previous review of this manuscript, the authors have performed a number of additional experiments to address my previous concerns, and are to be commended for undertaking these additional experiments. The manuscript is greatly improved as a result. I have a few additional concerns/queries below. My major concern is the use of penicillin in the Sp invasion experiments, as this antibiotic choice would also kill bacteria localised to the cytoplasm and damaged vacuoles. This raises the concern that the process being described is specific to antibiotic-killed bacteria.

Thank you for the positive comment.

1. (previous comment 2) In Fig 4, it seems a bit confusing to use two different colour schemes in Figs 4B and C. Suggest standardising the colour of the LC3 bars.

As suggested, we standardized the color of each marker. Please see the figures below.

2. (previous comment 2) In Fig 4B, I am confused as to why LC3, p62 and NDP52 was not included for both of panels 4B and 4C. If the model (in 4F) is correct, then NDP52 would stay associated with p62 staining in the FIP200-KO MEFs? For clarity, I think that P62 staining should be included in panel 4B, and NDP52 staining should be included in panel 4C.

Thank you for this insightful comment. As suggested by the reviewer, we examined the kinetics of NDP52 and p62 in the FIP200-KO MEFs and added this new data as Fig. 4D (Please see below, left panel). It clearly shows that the kinetics of NDP52- and p62-recruitment into pneumococci-containing vacuoles show distinct patterns. As described in the previous rebuttal letter, when either FIP200 or Atg14L is present, LC3 and NDP52 are removed from PCLV by 2 hours post-infection (p.i.). In other words, NDP52 stays

associated with p62 only when both FIP200 and Atg14L are absent. Therefore, in FIP200-KO cells (where Atg14L is present), NDP52 and LC3 are transiently recruited into pneumococci-containing vacuoles at 1 h p.i. and removed at 2 h p.i. To avoid misunderstanding, we corrected Figs 3J and 4I, and added a sentence to the revised manuscript (please see below (right panel) and lines 258-9).

Either
FIP200
or
Atg14L
Beclin1

3. (previous comment 4-3) Line 2 – should be “Upon knockdown of myosin VI”?

Thank you for this comment. As suggested, we have fixed it.

4. (previous comment 5) The use of penicillin G for bacterial invasion experiments to kill extracellular bacteria (lines 615-623) is a major concern, as penicillin efficiently enters the cell and kills bacteria in the cytoplasm and probably damaged endosomal compartments (see PMIDs 2501788 and 24331465, for example). This may account for the very low invasion rate (1-2%) seen in Fig S1Q. The major concern with this use of penicillin is that the autophagic process being described in this manuscript may be targeting penicillin-killed pneumococcus, rather than viable pneumococcus. To rule out

this possibility, this experiment should be repeated, but using gentamicin to kill extracellular bacteria.

Thank you for this insightful comment. We agree and understand your concern. We know that penicillin G is used for the investigation of endosomal membrane damage induced by some kinds of bacteria, including Shigella, Listeria, and GAS. As suggested by the reviewer, Walker's group treated cells with 100 µg/ml of Gentamicin (Gm) and 100 µg/ml of Penicillin G (PenG) for 2 h to kill cytosolic or cytosol-exposed bacteria (PMID 24331465; Cell Host Microbe. 2013, 14:675–682), and Portonoy's group treated cells with 10 µg/ml of Gentamicin (Gm) and 1000 µg/ml of methicillin for 24 h to kill cytosolic or cytosol-exposed Listeria monocytogenes (PMID 2501788; Proc Natl Acad Sci U S A. 1989, 86:5522-6.)

Before starting this study about the intracellular dynamism of S. pneumoniae, we carefully consulted previous reports to determine the conditions to kill extracellular S. pneumoniae. In the reviewed literature, Hammerschmidt's group treated cells with 100 µg/ml of Gm and 100 µg/ml of PenG for 1 h (J Biol Chem. 2010, 285:35615-23), Tuomanen's group treated cells with 200 µg/ml of Gm and 10 µg/ml of PenG for 1 h (J Clin Invest. 1998, 102:347-60), Paton's group treated cells with 200 µg/ml of Gm and 10 µg/ml of PenG for 1 h (Infect Immun. 1996, 64:3772–3777), and Blom's group treated cells with 100 µg/ml of Gm and 100 µg/ml of PenG for 1 h to kill extracellular S. pneumoniae (J Immunol. 2013, 191:4235-45). Furthermore, in a recent paper, Banerjee's group treated cells with 400 µg/ml of Gm and 10 µg/ml of PenG for 4 h to kill extracellular S. pneumoniae (PLoS Pathog. 2018, 14:e1007168).

Compared with the conditions used to kill cytosolic or cytosol-exposed GAS and Listeria, the conditions to kill extracellular S. pneumoniae used in these reports are modest from the point of PenG concentration and incubation period.

Furthermore, upon starting this study, we took PenG effect into consideration and carefully calibrated the Gm/PenG-killing conditions to evaluate intracellular burden of S. pneumoniae. As shown in the result below (left panel), we confirmed that the amount

of recovered S. pneumoniae from the cells treated with 100 µg/ml of Gm and 60 µg/ml of PenG for 1 h was level similar to that from the cells treated with 200 µg/ml of Gm for 20 min, indicating that 60 µg/ml of PenG for 1 h had no significant bactericidal effect on cytosolic or cytosol-exposed S. pneumoniae. After further calibration, we decided to treat cells as follows; medium including 200 µg/ml of Gm was added to the cells, which were then incubated for 15 min. The same amount of medium, including 200 µg/ml of Gm and 20 µg/ml of PenG (final 10 µg/ml), was added to the cells, and the cells were incubated for further 15 min.

Indeed, under these conditions, we confirmed that internalization of the adherence deficient mutant ($\Delta cbpA$ strain) was dramatically decreased, showing that killing of extracellular S. pneumoniae was sufficient (please see below, middle panel). Next, we confirmed that internalization of endosomal-damage deficient mutant (Δply strain) was similar to the level of wild type, showing that 10 µg/ml PenG treatment for 15 min had no significant reduction on the bacterial recovery from the cell (please see below, right panel).

We conclude that PenG shows no significant bactericidal effect on cytosolic or cytosol-exposed S. pneumoniae in our experimental setting, and we are convinced that the results in Fig. S1Q are reasonable.

5. (previous comment 7-1). I disagree with the notion that very few bacteria are invading into each cell, especially in light of my concern with using penicillin to kill extracellular bacteria (above). In multiple figures it can be observed that multiple bacteria have invaded individual cells (e.g. Fig 1A, at least 4 Sp invasions; Fig 1K, at least 5; Fig 3D, at least 7 for siLuc and 6 for Atg14L). My previous comment stands as a point of concern that should be addressed.

(previous comment 7-1). The method of quantitation of LC3-positive bacteria is not ideal in my opinion. With a very high MoI for these infections (MoI=100), it is quite likely that more than one bacteria will invade each cell.

As described in our response to comment 4, we carefully confirmed that PenG showed no significant bactericidal effect on cytosolic or cytosol-exposed S. pneumoniae in our experimental setting,

With regards to invasion efficiency, it was reported that some Streptococci, such as GAS MIT1 clone strain 5448, show high invasiveness, and this Streptococcal strain can infect cells at significantly low MOI (e.g., less than 1) (Cell Host Microbe. 2013, 14:675–682). Compared with this highly invasive GAS, the invasion efficiency of S. pneumoniae is quite low (0.07%-2% at MOI of 50) as referenced below.

1. MOI=50, 0.4-2 bacteria/cell (Infect Immun, 2005, 73:2680–9)

2. invasion efficiency 1% (Infect Immun, 1996,64:3772–7)

3. MOI=50, invasion efficiency 2% (J Biol Chem, 2009, 284:19427–36)

4. invasion efficiency 0.07-1 % (Cell, 2000, 102:827–37)

Therefore, due to its low invasiveness, a higher MOI is commonly used for infection of nonphagocytic cells with S. pneumoniae. In this study, the invasiveness of S. pneumoniae is around 1.5% at the MOI of 100, showing that our experimental setting is objectively reliable. Therefore, our MOI-setting is reasonable to achieve the conditions where all cells were invaded with at least one bacterial body. Under this carefully designed

experimental setting, we are convinced that counting marker-positive bacteria-containing cells is a reasonable and reliable method for the estimation of intracellular dynamics of S. pneumoniae.

6. (previous comment 7-2). I agree that counting chains could be challenging. Given that a chain of cocci would have arisen from a single invasion event, could each bacterial chain be counted as a single bacterium. i.e. quantitate as NDP52-pos chains etc.

As described in our previous rebuttal letter, intracellular pneumococci are observed as a chained coccus and frequently is seen in aggregate, like a cloud with various forms. As reviewer pointed out, we tried to count intracellular pneumococcal-chains and aggregates. However, we could not quantify marker-negative pneumococci to estimate the total number of pneumococci with reliability. However, we were able to count marker-positive pneumococci. Therefore, for the estimation of intracellular dynamics of pneumococci, we concluded that counting cells with marker-positive pneumococci was more reliable and reasonable.

Reviewer #3 (Remarks to the Author):

The authors have addressed most of my comments. However, I still have a minor comment/clarification:

Thank you for the positive comment.

1. In Fig. S7A,B the authors mention that "Notably, at 3 h p.i., LC3 recruitment remained high in siLuc-treated cells.

311 However, it was significantly decreased in siFIP200-treated cells, suggesting that subsequent PcAV is induced in siLuc-treated cells at 3 h p.i". But the data comparing the

siLuc-treated cells vs siFIP200-treated cells are on two different plots. Moreover, there is no statistical significance reported in the figures.

Thank you for this insightful comment. As suggested by reviewer, we added statistical tests to compare siLuc-treated cells to siFIP200-treated cells. To reflect this analysis, we added the information about LC3-recruitment at 4 h p.i. in siLuc treated cells (Please see below, right).

Reviewers' comments:

Reviewer #2 (Remarks to the Author):

All of my previous comments and concerns have been addressed, with the exception of the explanation for the use of penicillin (comment 4). The authors provide some nice background on the use of penicillin to kill extracellular bacteria in previous pneumococcal literature, but also provide their own experimental data that the addition of penicillin is not necessary to kill extracellular bacteria (graph showing invasion of WT Spn comparing Gm200 vs Gm100+Pen60 in their rebuttal letter).

However, my concern with the use of penicillin to kill extracellular bacteria is not related to invasion rates. In the early stages following invasion, the bacteria would be protected by the intact endosome (the rationale for the genetic screen in the referenced paper from Portnoy's lab). However, if there is escape into the cytoplasm and/or the endosomes are damaged (by PLO), then any penicillin that enters the cytosol could potentially be having an effect. It is notable that the data from the Portnoy lab's paper suggest that invasion rates in the presence of penicillin are not affected (Fig 1A), but survival does change (Fig1B vs A).

With regard to the levels of penicillin used and whether these are "modest", it really depends on the susceptibility of the strain being examined and the amount of penicillin that is able to penetrate into the cell. Most penicillin-sensitive clinical isolates have MICs less than 0.032ug/ml, which is ~300 times less than the concentrations being used in this study.

My suggestion is that the authors perform one additional experiment comparing Gm alone vs Gm+Pc (same concentrations used above) and measure viable cfu's and/or marker acquisition (e.g. LC3, NDP52) across a timecourse (0-3h). This data should be included as a supplementary figure.

Dear Reviewer:

We sincerely appreciate the helpful comments and suggestions provided by you with regard to our manuscript. We have read carefully all of your comments and have responded accordingly. Our detailed responses are mentioned below, including additional experimental data where appropriate. Revisions are highlighted in **blue** in the revised manuscript.

We hope that you find these revisions satisfactory.

Reviewer #2 (Remarks to the Author):

All of my previous comments and concerns have been addressed, with the exception of the explanation for the use of penicillin (comment 4). The authors provide some nice background on the use of penicillin to kill extracellular bacteria in previous pneumococcal literature, but also provide their own experimental data that the addition of penicillin is not necessary to kill extracellular bacteria (graph showing invasion of WT Spn comparing Gm200 vs Gm100+Pen60 in their rebuttal letter).

However, my concern with the use of penicillin to kill extracellular bacteria is not related to invasion rates. In the early stages following invasion, the bacteria would be protected by the intact endosome (the rationale for the genetic screen in the referenced paper from Portnoy's lab). However, if there is escape into the cytoplasm and/or the endosomes are damaged (by PLO), then any penicillin that enters the cytosol could potentially be having an effect. It is notable that the data from the Portnoy lab's paper suggest that invasion rates in the presence of penicillin are not affected (Fig 1A), but survival does change (Fig1B vs A).

With regard to the levels of penicillin used and whether these are "modest", it really depends on the susceptibility of the strain being examined and the amount of penicillin that is able to penetrate into the cell. Most penicillin-sensitive clinical isolates have MICs

less than 0.032ug/ml, which is ~300 times less that the concentrations being used in this study.

My suggestion is that the authors perform one additional experiment comparing Gm alone vs Gm+Pc (same concentrations used above) and measure viable cfu's and/or marker acquisition (e.g. LC3, NDP52) across a time course (0-3h). This data should be included as a supplementary figure.

Thank you for insightful comment. As suggested, we performed intracellular survivability assay of pneumococci under our experimental conditions and added this new data as Fig. S1R (Pease see below and line 159-61). It clearly shows that 10 µg/ml of penicillin G treatment for 15 min had no negative effect on intracellular survivability of pneumococci until 3 h, and that each cell was invaded by at least one bacterium in the presence or absence of penicillin G.

	Gm	Gm+PenG
Invasion efficiency (%)	0.86	0.92
Invaded bacteria/cell	1.10	1.18